# Acidity-activatable upconversion afterglow luminescence cocktail nanoparticles for ultrasensitive in vivo imaging

Yue Jiang[1,3], Min Zhao[1,3], Jia Miao[1], Wan Chen[1], Yuan Zhang[1], Minqian Miao[1], Li Yang[1], Qing Li[1] & Qingqing Miao ⓘ [1,2] ✉

Activatable afterglow luminescence nanoprobes enabling switched "off-on" signals in response to biomarkers have recently emerged to achieve reduced unspecific signals and improved imaging fidelity. However, such nanoprobes always use a biomarker-interrupted energy transfer to obtain an activatable signal, which necessitates a strict distance requisition between a donor and an acceptor moiety (<10 nm) and hence induces low efficiency and non-feasibility. Herein, we report organic upconversion afterglow luminescence cocktail nanoparticles (ALCNs) that instead utilize acidity-manipulated singlet oxygen ($^1O_2$) transfer between a donor and an acceptor moiety with enlarged distance and thus possess more efficiency and flexibility to achieve an activatable afterglow signal. After in vitro validation of acidity-activated afterglow luminescence, ALCNs achieve in vivo imaging of 4T1-xenograft subcutaneous tumors in female mice and orthotopic liver tumors in male mice with a high signal-to-noise ratio (SNR). As a representative targeting trial, Bio-ALCNs with biotin modification prove the enhanced targeting ability, sensitivity, and specificity for pulmonary metastasis and subcutaneous tumor imaging via systemic administration of nanoparticles in female mice, which also implies the potential broad utility of ALCNs for tumor imaging with diverse design flexibility. Therefore, this study provides an innovative and general approach for activatable afterglow imaging with better imaging performance than fluorescence imaging.

Optical imaging plays an indispensable role in biology and medicine by virtue of high sensitivity, non-invasiveness, and real-time longitudinal imaging[1–8]. Though the typically used fluorescence imaging is widely used for biomedical applications[9–17], it needs real-time light excitation, which inevitably induces spontaneous tissue autofluorescence and thus hampers imaging fidelity with a compromised signal-to-noise ratio (SNR)[18–20]. By contrast, self-luminescence imaging has attracted increasing attention as it detects photons without the need for real-

time light excitation and therefore eliminates autofluorescence, thus validating imaging with a high sensitivity and SNR[21,22]. So far, self-luminescence imaging mainly includes bioluminescence, chemiluminescence, Cherenkov luminescence, and afterglow luminescence imaging[23–25]. Among these techniques, bioluminescence imaging always necessitates specific gene transfection in living cells to catalyze the oxidation of substrates to release photons and hence the imaging sensitivity is usually affected by the cell environment and substrate

[1]State Key Laboratory of Radiation Medicine and Protection, School for Radiological and Interdisciplinary Sciences (RAD-X), Collaborative Innovation Center of Radiation Medicine of Jiangsu Higher Education Institutions, Soochow University, Suzhou 215123, China. [2]School of Nuclear Science and Technology, University of Science and Technology of China, Hefei 230026, China. [3]These authors contributed equally: Yue Jiang, Min Zhao. ✉e-mail: qqmiao@suda.edu.cn

availability[8,26–28]. Chemiluminescence imaging commonly relies on reactive species-initiated oxidation reactions to form the cyclic peroxides to trigger photonic release, which is easily perturbed by internal redox microenvironment[29–31]. Cherenkov luminescence imaging requires a high dosage of β-emitting radionuclides and the light-generation efficiency is relatively low[32,33]. In contrast, afterglow luminescence imaging is a process of capturing photonic energy in defects and slowly releasing photons after cessation of light illumination, which effectively eliminates autofluorescence and thus improves imaging sensitivity[34–38]. In addition, afterglow luminescence imaging is more flexible and wide-applicable as it does not require real-time light excitation, specific endoexogenous factors, or radioactive isotopes[39,40].

On account of the aforementioned merits, afterglow luminescence imaging particularly based on organic afterglow systems has become the research hotspot and is growing up as a powerful imaging technique. A few organic afterglow luminescence imaging probes including semiconducting polymers and small molecular systems have been utilized for diverse biomedical imaging applications[39,41–45]. However, most of the probes are designed to be "always on" with continuous signal output regardless of the proximity or binding to the molecular target of interest, thus compromising the sensing fidelity and real-time imaging capabilities of probes[46,47].

Thereby, researchers have explored activatable afterglow luminescence probes with switchable "off-on" signals in response to biomarkers, which can significantly reduce non-specific signals and thus improve imaging performance[48]. Up to now, only several activatable

afterglow luminescence probes have been designed for the detection of biothiols[36], H₂S[49,50], ONOO⁻[51–54], and protease[55]. The most common strategy is to design a biomarker-responsive quencher-modified linker as an energy acceptor to quench the afterglow luminescence that is restored upon biomarker-mediated cleavage of the quencher to release it from the nanoparticle surface[56,57]. However, due to the large distance between the afterglow donor and the quenching acceptor via nanoparticle surface modifications (>10 nm)[58,59], the energy transfer is relatively inefficient, thus hampering the quenching efficiency and thereafter the imaging contrast versus biomarkers. Therefore, an activatable afterglow approach that breaks such harsh distance limitation of energy transfer is urgently needed to expand versatility and improve sensing capability.

Herein, we report organic afterglow luminescence cocktail nanoparticles (ALCNs) for pH-activatable upconversion afterglow luminescence imaging (Fig. 1). ALCNs comprise a hybrid of afterglow initiator nanoparticles (AIN) as an "energy" donor to produce singlet oxygen ($^1O_2$) under NIR laser irradiation and afterglow substrate nanoparticles (ASN) as an "energy" acceptor to interact with $^1O_2$ to generate active dioxetane intermediates and subsequently release afterglow luminescence. Under physiological conditions, ALCNs are in the $^1O_2$ inactivated afterglow "off" state. In detail, both the donor AIN and acceptor ASN in ALCNs are negatively charged, which induces the separation of AIN and ASN via electrostatic repulsion. This blocks $^1O_2$ transfer from the AIN to ASN due to the limited transfer distance of $^1O_2$ in aqueous solution, preventing the downstream oxidation of afterglow substrate to form a dioxetane intermediate and generation of

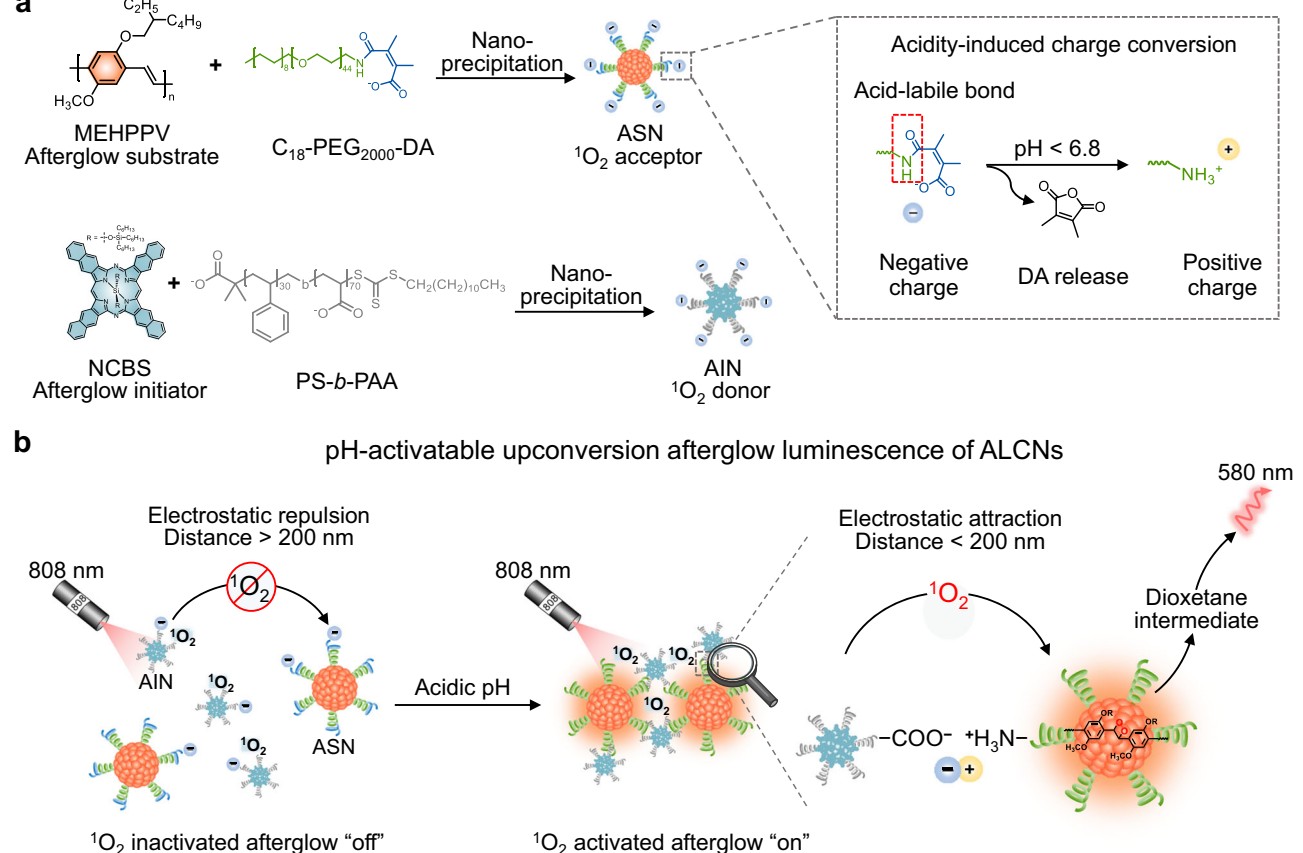

**Fig. 1 | Schematic illustration for the construction and mechanism of activatable afterglow luminescence cocktail nanoparticles (ALCNs) for tumor acidity imaging. a** Schematic illustration of the preparation of ASN and AIN through nanoprecipitation. **b** Schematic illustration of ALCNs for pH-activatable upconversion afterglow luminescence. Under physiological conditions, ALCNs are in the $^1O_2$ inactivated afterglow "off" state due to the electrostatic repulsion-mediated the block of $^1O_2$ transfer from the donor AIN to acceptor ASN. In contrast, acidic pH induces the charge reversal of ASN and mediates the proximity to AIN via electrostatic interaction, which facilitates $^1O_2$ transfer from AIN to ASN, ultimately inducing $^1O_2$ activated afterglow "on".

afterglow luminescence. In the acidic environment, the acid-sensitive amide bond of ASN is broken, which leads to its charge reversal from negative to positive charge on the nanoparticle surface. Thereby, the proximity between AIN and ASN, mediated by electrostatic interaction, enables efficient $^1O_2$ transfer from AIN to ASN, ultimately inducing $^1O_2$ activated afterglow "on". Beyond conventional energy transfer with a distance requisition within 10 nm between a donor and an acceptor, $^1O_2$ can be delivered within 200 nm in aqueous solution[60,61] and 10 - 55 nm in living cells[62-64], making feasibility and flexibility of the system for imaging usage. Besides, the activated afterglow luminescence intensity is linearly calibrated with the pH value. By virtue of minimized autofluorescence and acidity-specificity, ALCNs achieve in vivo imaging of 4T1-xenograft subcutaneous tumors and orthotopic liver tumors in living mice with a high SNR. Bio-ALCNs with biotin modification further demonstrate the wide-applicability and better imaging capability of tumor imaging via systemic administration.

## Results

### Synthesis and characterization of pH-activatable ALCNs

We began the study with chemical synthesis and nanoparticle fabrication of pH-sensitive ALCNs and pH-inert control cocktail nanoparticles C-ALCNs (Fig. 1 and Supplementary Fig. 1). Firstly, 2, 3-dimethylmaleic anhydride (DA) and succinic anhydride (SA) were reacted with $C_{18}$-$PEG_{2000}$-$NH_2$ to obtain pH-responsive and pH-inert amphiphilic polymers (i.e., $C_{18}$-$PEG_{2000}$-DA and $C_{18}$-$PEG_{2000}$-SA), respectively (Supplementary Fig. 2). Then, the polymeric afterglow substrate poly[2-methoxy-5-(2′-ethylhexyloxy)−1,4-phenylene vinylene] (MEHPPV) serving as a $^1O_2$ acceptor moiety was encapsulated with $C_{18}$-$PEG_{2000}$-DA and $C_{18}$-$PEG_{2000}$-SA via a nanoprecipitation method to get water-dispersible ASN and C-ASN nanoparticles, respectively (Fig. 1 and Supplementary Fig. 1). Similarly, the negative-charged nanoparticles were prepared by co-precipitating the $^1O_2$ photosensitizer silicon 2,3-naphthalocyanine bis(trihexylsilyloxide) (NCBS) with the carboxyl-terminated amphiphilic polymer polystyrene-block-poly(acrylic acid) (PS-b-PAA).

As follows, the size distribution and morphology of the obtained nanoparticles were investigated (Fig. 2a, b and Supplementary Fig. 3). Dynamic light scattering (DLS) analyses indicated ASN and C-ASN had a comparable average hydrodynamic size, which was $79.7 \pm 2.3$ nm and $81.3 \pm 1.9$ nm, respectively (Fig. 2a and Supplementary Fig. 3). Differently, AIN had a distinctly smaller average hydrodynamic size of $20.6 \pm 1.6$ nm due to the small molecular NCBS as the hydrophobic core to form nanoparticles relative to a polymeric hydrophobic core of ASN and C-ASN (Fig. 2b). Transmission electron microscopy (TEM) further revealed a spherical morphology of the obtained nanoparticles (Fig. 2a, b).

To evaluate the pH-dependent charge reversal capability of ASN, the nanoparticles were incubated with buffered solution at different pH. As shown in Fig. 2c, the zeta potential of ASN changed rapidly from $−21.9 \pm 2.3$ mV to $2.3 \pm 0.25$ mV after incubation with buffered solution at pH 5.5 for 20 min. After 6 h of continuous incubation, the zeta potential of ASN gradually increased and maintained a plateau, implying the complete hydrolysis of the acid-labile amide bond to liberate an amino-termini group with a positive charge on the surface of the nanoparticles. Besides, the pH-sensitive charge conversion of ASN was also observed at pH 6.5 (Fig. 2c). In contrast, no charge conversion took place for pH-insensitive C-ASN at any pH values (Supplementary Fig. 4a). In addition, PS-b-PAA-coated AIN nanoparticles showed a negative charge no matter at alkaline or acidic pH, indicating their ideal stability as the negative-charged $^1O_2$ donor nanoparticles (Supplementary Fig. 4b). These results suggested that the surface charge of ASN could change from negative to remarkably positive in acidity, thus allowing effective electrostatic interaction between ASN and AIN for $^1O_2$ transfer within ALCNs.

To further validate the acidity-induced amplified upconversion afterglow luminescence, the optical properties of ALCNs and C-ALCNs at different pH were recorded and compared (Fig. 2d–i and Supplementary Figs. 5, 6). ALCNs and C-ALCNs showed consistent absorption spectra with absorption peaks at 494 nm and 783 nm at different pH values, which originated from individual $^1O_2$-acceptor ASN or C-ASN and donor AIN, respectively (Fig. 2d and Supplementary Fig. 5). Differently, ALCNs presented unchanged fluorescence peak at 590 nm but a discernably enhanced fluorescence peak at 793 nm with decreasing pH from 7.4 to 5.5 (Fig. 2e). Though such emerged fluorescence peak at 793 nm derived from NCBS was tiny, it verified an occurrence of fluorescence resonance energy transfer (FRET) from MEHPPV to NCBS, which was attributed to the electrostatic attraction-mediated proximity between ASN and AIN in an acidic environment (Supplementary Fig. 6). Meantime, an obvious size increase of ALCNs was observed with decreasing the pH values, further demonstrating the electrostatic attraction-mediated proximity and aggregation of ASN and AIN in acidic conditions (Supplementary Fig. 7). As expected, no emission peak at 793 nm was observed for C-ALCNs in the same pH range due to its pH-insensitive properties and thus no occurrence of electrostatic attraction (Supplementary Fig. 6d). Next, the afterglow luminescence signals of ALCNs and C-ALCNs were collected with the IVIS Spectrum imaging system after an 808 nm laser or white light pre-irradiation for 1 min (Fig. 2f–i). Under an 808 nm laser pre-irradiation, a remarkable enhancement of the afterglow luminescence signal was generated for ALCNs with decreasing pH values, which was enhanced by 11.3 folds from pH 7.4 to 5.0 (Fig. 2h, i). The activated afterglow luminescence possessed a half-life of 2 min (Supplementary Fig. 8). Note that an 808 nm laser could selectively sensitize AIN to generate $^1O_2$, implying the successful $^1O_2$ transfer from AIN to ASN to activate the afterglow signal of ALCNs via acidity-induced electrostatic attraction between AIN and ASN. In contrast, there was no observable afterglow luminescence signal for C-ALCNs under the same conditions because they are pH-insensitive and cannot mediate the nanoparticle interaction for $^1O_2$ transfer and afterglow brightness. In addition, the afterglow spectra of ALCNs at different pH were consistent with fluorescence spectra (Fig. 2f). To further verify that the afterglow brightness was activated by the transfer of $^1O_2$ from AIN to ASN under pre-irradiation with an 808 nm laser, afterglow imaging under white light pre-irradiation was also performed. As expected, a weak but no significant change in the afterglow luminescence signals of both nanoparticles was observed after white light pre-irradiation at all pH values tested (Fig. 2h, i). This was because both ASN and C-ASN in the respective ALCNs and C-ALCNs could absorb visible light to self-sensitize to generate afterglow luminescence and were therefore irrelevant to pH. In addition, the ratio of ASN and AIN within the ALCNs, laser irradiation time and power density were optimized and a concentration-dependent afterglow signal was observed (Supplementary Fig. 9). The afterglow half-life changed conversely with the afterglow intensity of ALCNs at different laser irradiation time, power density and the nanoparticle concentrations, which was probably relevant to the duration time of unstable peroxides at different conditions (Supplementary Fig. 10). Besides, a linear correlation relationship between the afterglow intensity of ASN under 808 nm laser pre-irradiation and the pH values was obtained, indicating the ideal quantitative performance (Fig. 2j).

### In vitro activatable afterglow luminescence imaging of acidic pH

Before investigating the acid-induced activated afterglow luminescence of ALCNs towards cancer cells, a cytotoxic study was carried out via CCK-8 assays (Supplementary Fig. 11). The results proved that ASN, C-ASN, and AIN showed no obvious cytotoxicity to cells. To test the response of probes towards an acidic environment in living cells, ALCNs and C-ALCNs were incubated with HepG2 cells for 24 h, followed by incubation with culture medium with or without pH

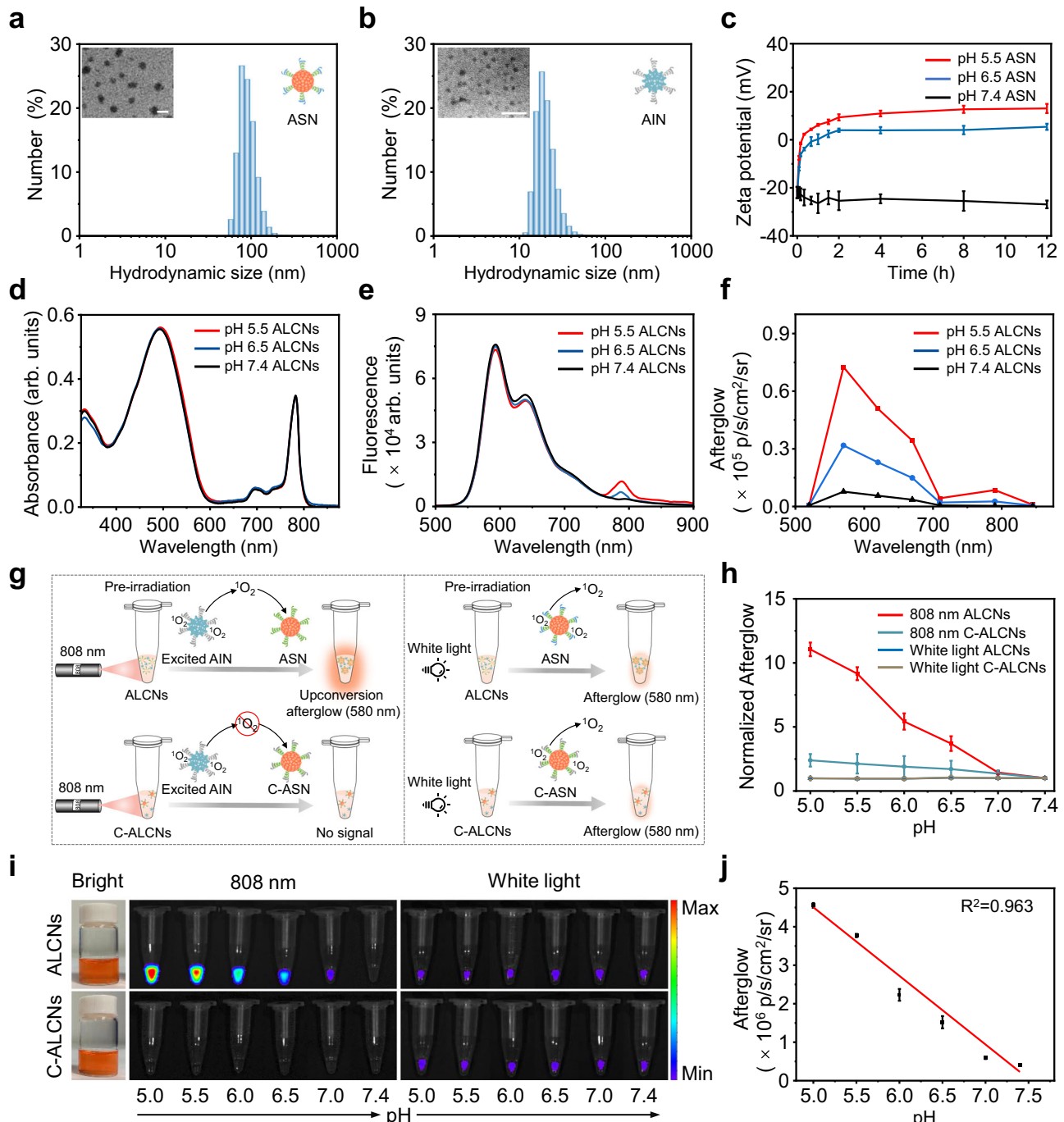

**Fig. 2 | Characterization and study of the pH-responsive activatable upconversion afterglow luminescence of ALCNs.** Average hydrodynamic diameters and representative TEM images of ASN (**a**), and AIN (**b**) in 1 × HEPES buffer (pH = 7.4) (the scale bar represents 100 nm). **c** Time-dependent zeta potential changes of ASN (10 μg/mL) at different pH values in 1 × HEPES buffer ($n = 3$ independent experiments). The zeta potential values of pH 5.5 and 6.5 at 0 h were based on that of pH 7.4. **d** UV-Vis absorption spectra of ALCNs (10 μg/mL ASN, and 1 μg/mL AIN) at different pH values in 1 × HEPES buffer. **e** Fluorescence spectra of ALCNs (10 μg/mL ASN, and 1 μg/mL AIN) at different pH values in 1 × HEPES buffer with an excitation wavelength at 480 nm. **f** Afterglow spectra of ALCNs (10 μg/mL ASN, and 1 μg/mL AIN) at different pH values in 1 × HEPES buffer. **g** Schematic illustration of afterglow luminescence generation from ALCNs or C-ALCNs after pre-irradiation with an 808 nm laser or white light for 1 min. **h** Normalized afterglow intensities of ALCNs or C-ALCNs (20 μg/mL ASN or C-ASN, and 2 μg/mL AIN) under 808 nm laser and white light pre-irradiation ($n = 3$ independent experiments). **i** Bright-field (left panel), afterglow luminescence under 808 nm laser pre-irradiation (middle panel) and under white light pre-irradiation (right panel) images of ALCNs or C-ALCNs (20 μg/mL ASN or C-ASN, and 2 μg/mL AIN) at different pH values in 1× HEPES buffer. The afterglow images were captured after ALCNs or C-ALCNs were illuminated with light beam generated by 808 nm laser (1 W/cm²) and white light (0.1 W/cm²) for 1 min respectively, on an IVIS Spectrum imaging system. **j** Fitted calibration curve for the afterglow intensities of ALCNs as a function of pH values ($n = 3$ independent experiments). Data are presented as mean ± s.d. Source data are provided as a Source Data file.

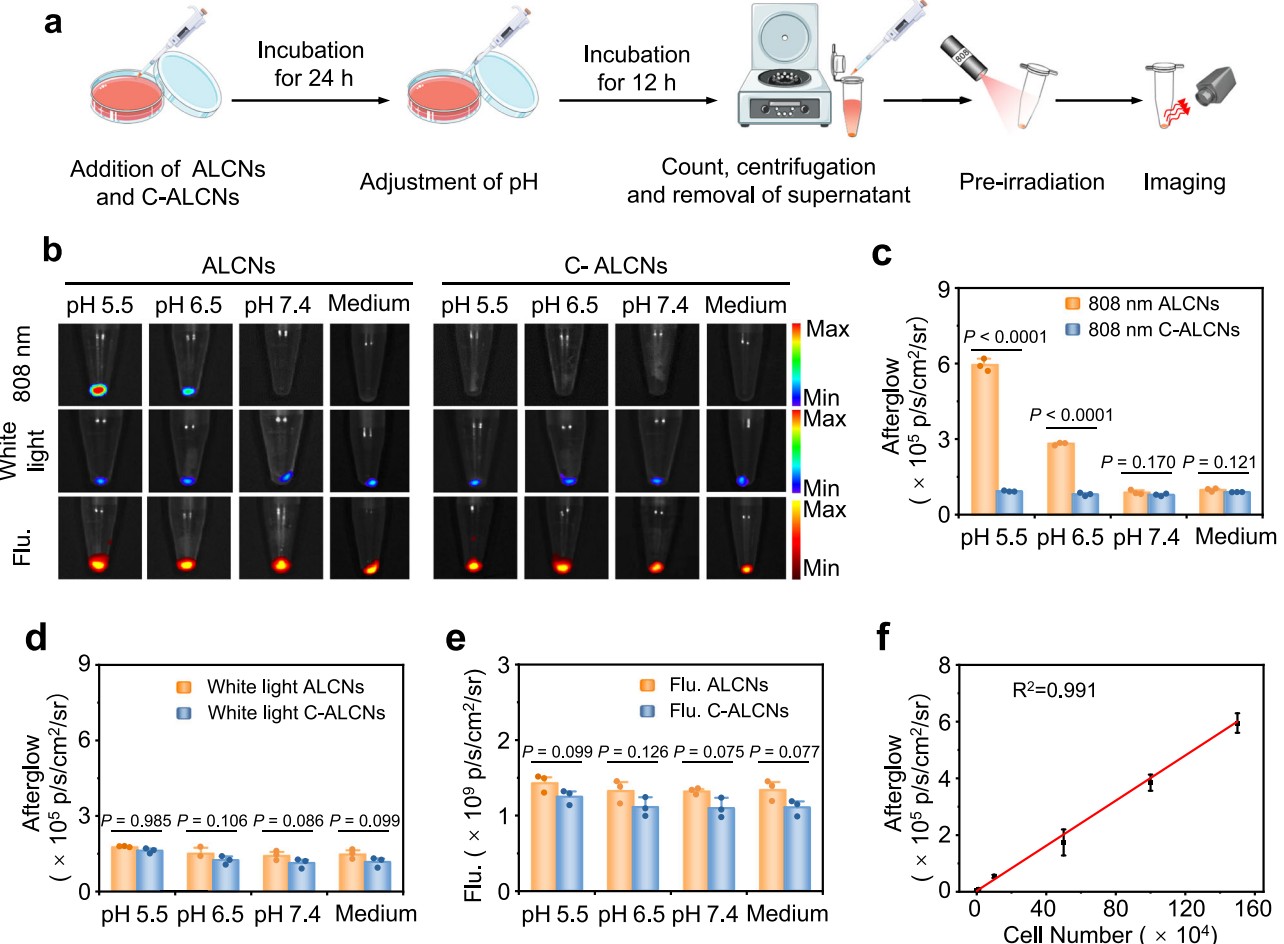

**Fig. 3 | In vitro afterglow luminescence imaging capability of ALCNs towards HepG2 cells. a** Schematic illustration of the detailed procedures for afterglow luminescence imaging of HepG2 cells. **b** Afterglow luminescence images of cell pellets (about $4 \times 10^6$ HepG2 cells per cell dish) after incubation with ALCNs or C-ALCNs (30 μg/mL ASN or C-ASN, and 3 μg/mL AIN) at different pH values and without pH adjustment under 808 nm pre-irradiation (upper panel) or white light pre-irradiation (middle panel), and corresponding fluorescence images (bottom panel). The fluorescence images were acquired with an excitation at 480 nm and emission wavelength at 580 nm. Quantification of afterglow intensities under 808 nm laser pre-irradiation (**c**), under white light pre-irradiation (**d**), fluorescence intensities (**e**) of HepG2 cells incubation of ALCNs and C-ALCNs. **f** Linear fitting curve of afterglow intensities with different cell numbers under 808 nm laser pre-irradiation of ALCNs in 1 × HEPES buffer (pH = 5.5). Data are presented as mean ± s.d. and analyzed by the Student's two-sided *t* test (*n* = 3 biologically independent samples). Source data are provided as a Source Data file.

adjustment for 12 h. After incubation, the underlying cells were counted, centrifuged, and harvested for afterglow luminescence imaging on the IVIS Spectrum imaging system (Fig. 3a). The cells after incubation with ALCNs obviously showed a brighter afterglow luminescence signal with decreasing incubation pH after an 808 nm laser pre-irradiation, which was increased by 6.0 folds at pH 5.5 relative to that at 7.4 (Fig. 3b, c). No obviously activated afterglow luminescence in the groups after incubation with medium or pH 7.4 was observed, indicating extracellularly acidic microenvironment indeed facilitates the charge converse and nanoparticle proximity for activated afterglow signal in tumor cells. In comparison, there was no significant difference in afterglow luminescence signals for C-ALCNs-treated HepG2 cells at all tested pH values. Besides, similar to the in vitro study in Fig. 2i, there was no significant difference in the afterglow luminescence intensity of each group after white light pre-irradiation (Fig. 3d). Also, fluorescence intensity at 580 nm derived from ASN presented no obvious changes among groups at different pH (Fig. 3e). Remarkably, for both HepG2 and 4T1 cells, the activated afterglow luminescence intensities of ALCNs could be linearly correlated with the cell numbers (Fig. 3f and Supplementary Figs. 12–14). As the emission of NCBS at 775 nm can't be detected by the confocal microscope, R-AIN was prepared with doped alkylated rose bengal (a-RB) to monitor its subcellular colocalization

(Supplementary Fig. 15). Co-localization study showed that ASN and R-AIN colocalized mainly in lysosomes (Supplementary Fig. 16). All the results demonstrated that ALCNs could activate upconversion afterglow luminescence in living cells in the acidity with high specificity and quantification capability.

### Afterglow imaging of subcutaneous tumors in mice

To explore the feasibility of ALCNs for in vivo imaging of an acidic tumor environment, 4T1 tumor-bearing xenograft mice were applied for activatable afterglow luminescence imaging. ALCNs and acidity-insensitive control C-ALCNs were injected intratumorally (i.t.) or intramuscularly (i.m.), and their fluorescence and afterglow luminescence signals under an 808 nm laser or white light pre-irradiation were recorded over time (Fig. 4a). As shown in Fig. 4b, the mice treated with ALCNs at the muscle site and the mice treated with C-ALCNs no matter in the tumor or muscle site after an 808 nm laser pre-irradiation showed a minimal afterglow luminescence signal at all the time points. By contrast, an intense afterglow luminescence signal was observed at the tumor site of mice treated with ALCNs at 1 h post-injection. The afterglow intensities reached a maximum with an SNR of 632.3 at 4 h post-injection, which was 8.1-fold, 7.0-fold, and 9.3-fold than those of the mice treated with ALCNs at the muscle site, the mice treated with

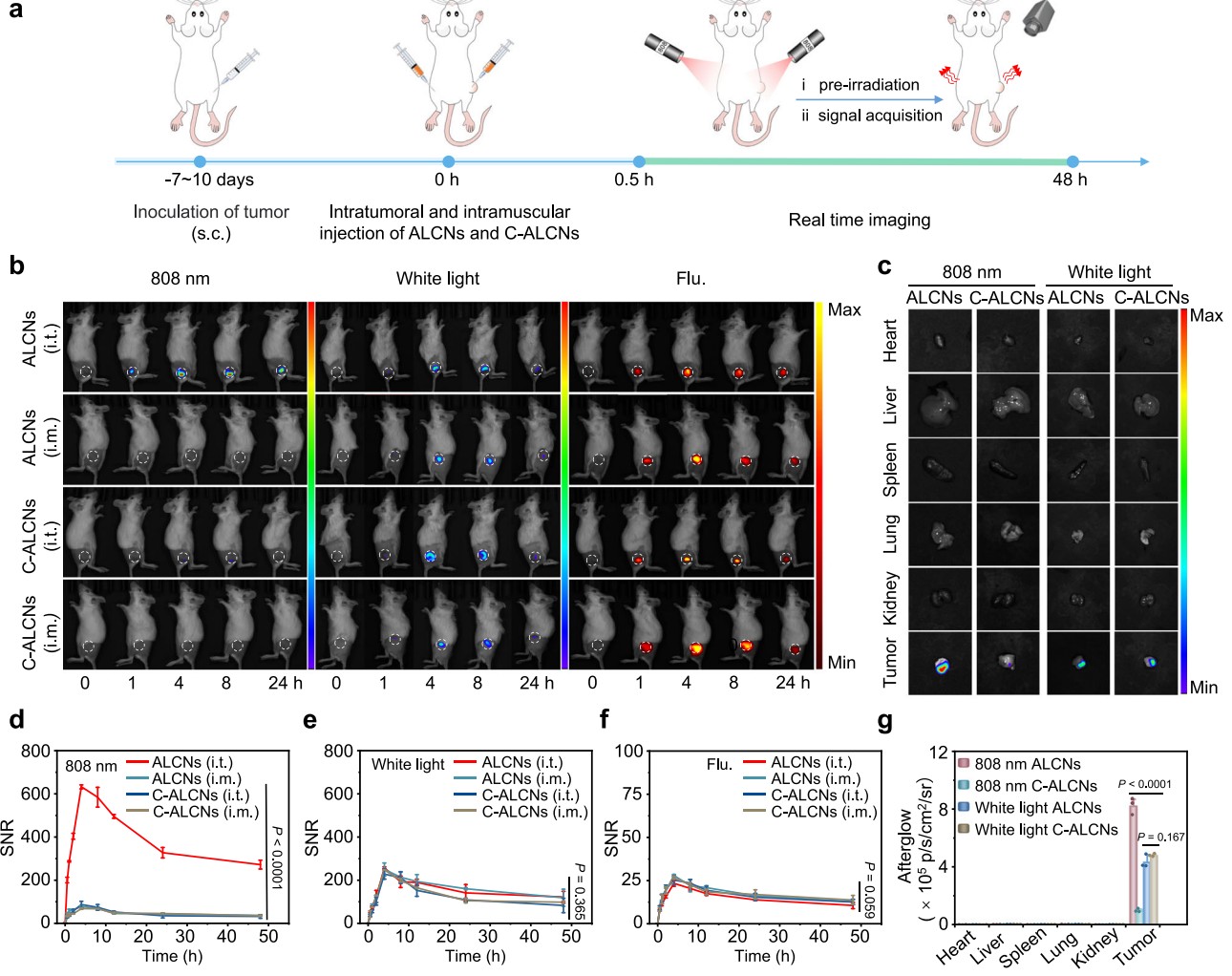

**Fig. 4 | In vivo afterglow luminescence imaging of 4T1-xenograft tumors.**
**a** Schematic illustration of the detailed procedures of 4T1 tumors through the afterglow imaging of ALCNs and C-ALCNs. **b** Afterglow images under an 808 nm pre-irradiation (left panel), white light pre-irradiation (middle panel), and fluorescence images (right panel) after intratumor and intramuscular injection of ALCNs or C-ALCNs (1 mg/kg ASN or C-ASN, and 0.1 mg/kg AIN). The fluorescence images were acquired with an excitation wavelength at 480 nm and emission wavelength at 580 nm. **c** Ex vivo afterglow images of various tissues under an 808 nm and white light pre-irradiation from 4T1-xenograft subcutaneous tumor-bearing mice at 48 h post-injection of ALCNs or C-ALCNs. The quantified SNRs for afterglow luminescence imaging under an 808 nm pre-irradiation (**d**), white light pre-irradiation (**e**), fluorescence luminescence imaging (**f**) of 4T1-xenograft subcutaneous tumor-bearing mice as a function of time. **g** Quantification of ex vivo afterglow intensities of various tissues in (**c**). Data are presented as mean ± s.d. and analyzed by one-way ANOVA (*n* = 3 mice each group). Source data are provided as a Source Data file.

C-ALCNs at the tumor site, and the mice treated with C-ALCNs at the muscle site, respectively (Fig. 4d). Attributing to the low background of afterglow luminescence, such acidity-activatable afterglow luminescence signal could last long for 48 h with an SNR of up to 272.2, indicating its suitability for in vivo long-term real-time imaging. However, it was unachievable for ALCNs with afterglow imaging after white light pre-irradiation or with fluorescence imaging to clearly discern tumors from normal tissues, highlighting the ability of ALCNs for in vivo pH imaging in upconversion mode (Fig. 4e, f). These results were consistent with ex vivo afterglow luminescence and fluorescence imaging of resected tumor tissue (Fig. 4c, g and Supplementary Fig. 17), confirming that ALCNs can detect and respond to the acidic tumor microenvironment (TME) of 4T1 tumors. Moreover, the fluorescence images of tumor sections demonstrated obvious co-localization of ASN and AIN both intracellularly and extracellularly, implying the high possibility of nanoparticle proximity for ¹O₂ transfer and activatable afterglow luminescence of ALCNs (Supplementary Fig. 18). The acidic microenvironment-activatable upconversion afterglow luminescence was also observed in HepG2-tumor-bearing

mice, verifying the general utility of ALCNs for tumor imaging (Supplementary Figs. 19, 20).

**Afterglow imaging of orthotopic liver tumors**
By virtue of high sensitivity and acidity-activated afterglow imaging capability, ALCNs were further used to detect orthotopic liver tumor in living mice. Before in vivo imaging, the pharmacokinetics of AIN and ASN were investigated (Supplementary Fig. 21). By endowing different nanoparticle sizes, ASN and AIN established a different plasma half-life of 11.9 and 33.6 min, respectively. The afterglow luminescence via an 808 nm laser or white light pre-irradiation and fluorescence signals were continuously acquired from 0 to 48 h after tail vein injection (Fig. 5a). As shown in Fig. 5b, the mice injected with ALCNs showed a discernable afterglow signal in the liver site by an 808 nm laser pre-irradiation at 1 h and gradually increased over time. At 4 h post-injection of ALCNs, the afterglow signal reached the highest, which was 3.6 times higher than that of the pH-insensitive C-ALCNs probe under upconversion mode at this time point (Fig. 5b, c). At 4 h post-injection, the suspected lesion could also be visualized under white light pre-

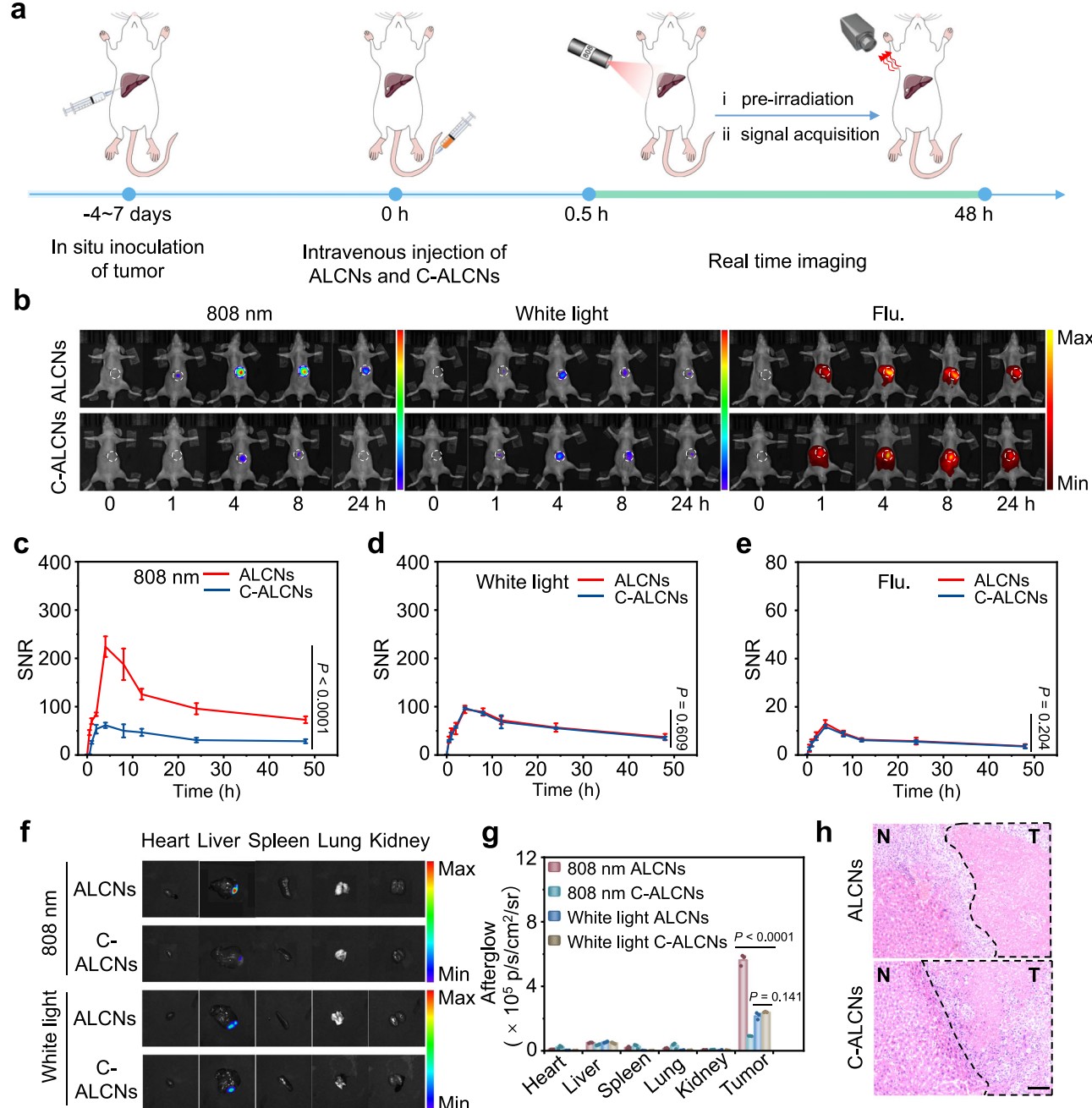

**Fig. 5 | In vivo activatable afterglow imaging of orthotopic liver tumors.**
**a** Schematic illustration of the detailed procedures of hepatocellular carcinoma in situ through the afterglow imaging. **b** Afterglow images under an 808 nm pre-irradiation (left panel), under white light pre-irradiation (middle panel) and fluorescence images (right panel) after intravenous injection of ALCNs or C-ALCNs (4 mg/kg ASN or C-ASN, and 0.4 mg/kg AIN). The fluorescence images were acquired with an excitation wavelength at 480 nm and emission wavelength at 580 nm. The quantified SNRs for afterglow luminescence imaging under 808 nm pre-irradiation (**c**), under white light pre-irradiation (**d**), fluorescence imaging (**e**) of orthotopic liver tumor-bearing mice as a function of time. **f** Ex vivo afterglow images of various tissues under an 808 nm and white light pre-irradiation from an orthotopic liver tumor-bearing mice at 48 h post- injection of ALCNs or C-ALCNs intravenously. **g** Quantification of ex vivo afterglow intensities of various tissues in (**f**). **h** H&E staining images of tumor slices of mice after injection with ALCNs or C-ALCNs (N: normal tissue, T: tumor tissue). The scale bar represents 100 µm. Data are presented as mean ± s.d. and analyzed by the one-way ANOVA (*n* = 3 mice each group). Source data are provided as a Source Data file.

irradiation as well as fluorescence imaging in both ALCNs and C-ALCNs-treated mice. This was caused by the nanoparticle accumulation in tumors via the enhanced permeability retention (EPR) effect of nanoparticles, which allowed for signal collection of an ASN component within ALCNs and C-ALCNs under white light pre-irradiation or fluorescence imaging (Fig. 5b). Nevertheless, attributing to tumor acidity-mediated afterglow amplification, high sensitivity, and high tissue penetration depth of 808 nm light, ALCNs displayed the highest

imaging SNR in upconversion mode, which was 2.3 folds and 17.2 folds higher than that under white light pre-irradiation or fluorescence imaging at 4 h post injection of ALCNs (Fig. 5c–e). As compared with most of previously reported organic afterglow system (Supplementary Table 1), ALCNs endow a comparable SNR and high specificity for disease-associated in vivo imaging.

The afterglow luminescence and fluorescence images of various organs in mice for biodistribution study after treatment with ALCNs or

C-ALCNs for 48 h were obtained and the signal intensities were quantified (Fig. 5f, g and Supplementary Fig. 22). The liver from mice treated with ALCNs and 808 nm laser pre-irradiation presented the highest afterglow signal relative to other control groups, which was consistent with in vivo results. Besides, the tumor lesions resided in the liver with signal were confirmed by histological hematoxylin and eosin (H&E) staining, confirming the capability of ALCNs for specific detection of orthotopic tumors (Fig. 5h). The biodistribution study of ASN and AIN after systemic administration at 4 and 24 h post-injection of nanoparticles was also performed. As shown in Supplementary Fig. 23, both of ASN and AIN established the highest accumulation capacity in the liver tumor site owing to the hepatobiliary metabolism pathway and EPR effect of nanoparticles, which further demonstrated the feasibility of ALCNs for orthotopic liver tumor imaging. Clear co-localization of ASN and AIN was also seen in the fluorescence images of tumor section, revealing the high tumor targeting and high potentiality of nanoparticle proximity for afterglow activation after systemic administration (Supplementary Fig. 18). The biocompatibility of nanoparticles was validated by H&E staining of various organs of the mice after treatment with ALCNs or C-ALCNs for 48 h (Supplementary Fig. 24). Taken together, these results indicated that ALCNs can detect orthotopic liver tumors with high sensitivity, specificity and efficiency.

### Afterglow imaging of tumors with Bio-ALCNs

To further enhance the tumor targeting ability of ALCNs for exploring the wide applicability, a tumor targeted group (i.e., biotin) was modified on the nanoparticles to obtain Bio-ALCNs (Fig. 6a). Compared with ALCNs, Bio-ALCNs possessed comparable size distribution and consistent afterglow responsivity towards acidic pH (Supplementary Fig. 25). Bio-ALCNs were firstly tested in biotin receptor-overexpressed B16F10 tumor-bearing xenograft mice[65,66]. After systemic administration of Bio-ALCNs, a gradually enhanced afterglow signal was observed, which reached the maximum at 12 h post-injection of nanoparticles (Supplementary Fig. 26). At this time point, the afterglow SNR of Bio-ALCNs-treated mice was 2.5- and 9.5-fold higher than that of ALCNs- and C-ALCNs-treated mice, showing the high tumor targeting and responsivity of Bio-ALCNs (Supplementary Fig. 26). Fluorescence imaging also demonstrated the high tumor accumulation capacity of nanoparticles after biotin modification, but endowing obviously decreased SNR due to its high autofluorescence (Supplementary Fig. 26). Ex vivo imaging of resected tumor tissues, H&E analysis, and nanoparticles co-localization via fluorescence imaging of tumor sections further demonstrated the activated afterglow signal was actually originated from the activatable afterglow signal of Bio-ALCNs resided in the tumor tissues (Supplementary Figs. 26–28). The higher fluorescence signal in the liver relative to the other groups was probably derived from the metabolization of nanoparticles in the tumors via the liver tissue (Supplementary Fig. 28).

After demonstrating the imaging capability of subcutaneous xenograft tumor model via systemic administration, Bio-ALCNs were further investigated for imaging of pulmonary metastases in living mice (Fig. 6). The afterglow luminescence and fluorescence signals were continuously acquired from 0 to 48 h after tail vein injection of nanoparticles (Fig. 6a). As shown in Fig. 6b, an obvious afterglow signal in the lung site emerged at 8 h and reached the peak at 24 h post-injection of Bio-ALCNs. Owing to the enhanced targeting efficiency after biotin modification, the SNR of Bio-ALCNs was as high as 410.0, which was 3.7-fold higher than that of ALCNs (Fig. 6b, d). In comparison, an ignorable afterglow signal was observed in the acidity-insensitive C-ALCNs-treated control group, confirming the pH-responsivity ability of Bio-ALCNs. Surprisingly, at all the observable time windows, the suspected lesion in the lung site could not be clearly delineated in fluorescence imaging. The data confirmed the ultrahigh sensitivity and specificity of pH-

activatable Bio-ALCNs for imaging of pulmonary metastases in afterglow imaging as compared with fluorescence imaging (Fig. 6d, e). Biodistribution study showed consistent results even at 48 h post-injection of nanoparticles (Fig. 6c, f and Supplementary Fig. 29). Moreover, the H&E assay and obvious co-localization of ASN and AIN through fluorescence imaging of the suspected lesion further supported the specific detection of pulmonary metastatic tumors by Bio-ALCNs (Fig. 6g and Supplementary Fig. 30). Taken together, as a representative trial, this result proved the great targeting ability, sensitivity, and specificity of Bio-ALCNs for tumor imaging and also implied the potential broad utility of ALCNs for tumor detection with diverse design flexibility.

## Discussion

In summary, we have developed ALCNs with acidity-activatable upconversion afterglow luminescence for tumor imaging. ALCNs utilized acidic pH-controlled $^1O_2$ transfer from donor to acceptor with a longer required distance relative to conventional energy transfer necessitating a distance between a donor and acceptor within 10 nm, which was more feasible and efficient. In comparison to the acid-insensitive control (i.e., C-ALCNs), ALCNs generated acidity-responsive activatable upconversion afterglow luminescence intensities upon an 808 nm laser pre-irradiation via pH-controlled proximity of donor and acceptor nanoparticles to achieve $^1O_2$ transfer from AIN to ASN within nanoparticles. Such acidity-mediated $^1O_2$-transfer activated afterglow luminescence of ALCNs was further validated in living cells. By virtue of acidity-mediated activatable upconversion afterglow luminescence and minimized autofluorescence, ALCNs achieved discriminative detection between 4T1-xenograft subcutaneous tumor and normal tissue with an 808 nm laser pre-irradiation, while afterglow luminescence imaging under white light pre-irradiation, fluorescence imaging and non-responsive control probe failed to do it. Moreover, systemic administration of ALCNs allowed for in vivo imaging of orthotopic liver tumors in living mice with a high SNR relative to fluorescence imaging. Advanced Bio-ALCNs achieved enhanced targeting ability and imaging performance for tumor imaging via systemic administration, indicating the potential broad utility of ALCNs for tumor detection with diverse design flexibility. Together, this study provides an effective and general platform for designing activatable afterglow luminescence probes.

The acid-labile amide bond of ASN within ALCNs is hydrolyzed to liberate the amino-termini group in an acidic environment, which is an irreversible process. This induces the "off-on" afterglow being easily perturbed by the tumor environment and thus is not applicable in a complex and dynamic biological environment. Therefore, advances of acid-activatable afterglow luminescence from irreversibility to reversibility will be further explored in the future study. Given that the donor and acceptor nanoparticles within ALCNs possess inconsistent physiological circulation and biodistribution, the gathering efficiency of nanoparticles within tumors is low, resulting in compromised imaging performance[67]. Though Bio-ALCNs with enhanced tumor targeting ability present better imaging capacity relative to ALCNs, it can be only adapted to the tumors with overexpressed biotin receptor[68,69]. Therefore, it is deserved to develop new strategies in terms of precise nanoengineering of nanoparticles to increase the tumor accumulation efficiency of the nanoparticles within diseased sites as well as broaden the applicability in different diseases[70,71].

## Methods

### Ethics statements

All animal experiments conducted in this study were carried out following the guidelines established by the Care and Use of Laboratory Animals of Soochow University and were approved by the Animal Ethics Committee of the Soochow University Laboratory Animal Center (Suzhou, China, 202110A0067).

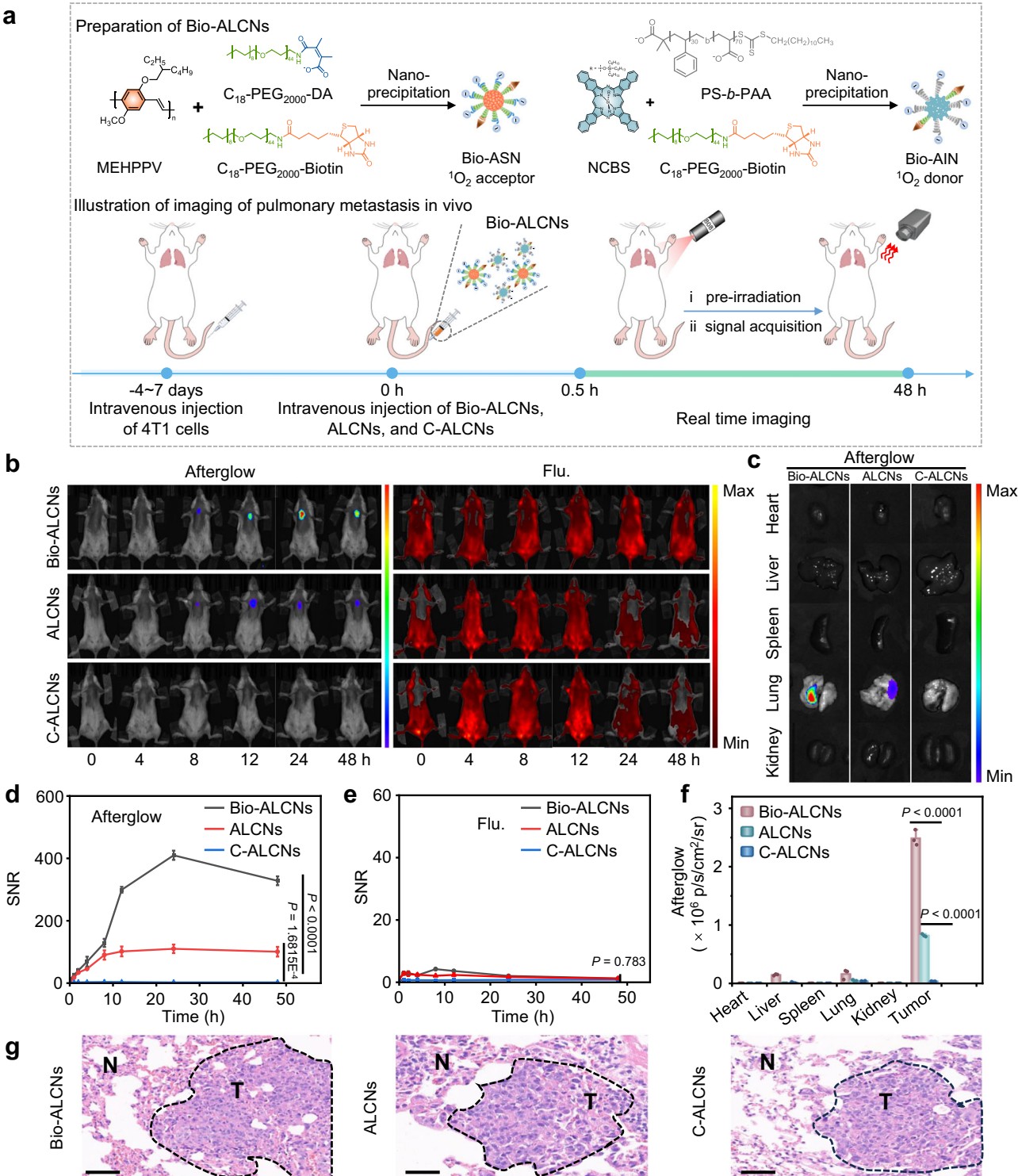

**Fig. 6 | In vivo activatable afterglow imaging of pulmonary metastasis tumors.**
**a** Schematic illustration of the detailed procedures of afterglow imaging of 4T1
pulmonary metastasis tumors. **b** Afterglow images under an 808 nm pre-irradiation
(left panel), and fluorescence images (right panel) after intravenous injection of
Bio-ALCNs, ALCNs or C-ALCNs (4 mg/kg Bio-ASN, ASN or C-ASN, and 0.4 mg/kg Bio-
AIN, or AIN). The fluorescence images were acquired with an excitation wavelength
at 480 nm and emission wavelength at 580 nm. **c** Ex vivo afterglow images of var-
ious tissues under an 808 nm laser pre-irradiation from 4T1 pulmonary metastasis-
bearing mice at 48 h post-injection of Bio-ALCNs, ALCNs or C-ALCNs intravenously.

The quantified SNRs for afterglow luminescence imaging under 808 nm pre-
irradiation (**d**), fluorescence imaging (**e**) of 4T1 pulmonary metastasis tumor-
bearing mice as a function of time. **f** Quantification of ex vivo afterglow intensities
of various tissues in (**c**). **g** H&E staining images of tumor slices of mice after treat-
ment with Bio-ALCNs, ALCNs or C-ALCNs (N: normal tissue, T: tumor tissue). The
scale bar represents 100 μm. Data are presented as mean ± s.d. and analyzed by the
one-way ANOVA (n = 3 mice each group). Source data are provided as a Source
Data file.

## Chemical synthesis and characterization

Synthesis of $C_{18}$-$PEG_{2000}$-DA. $C_{18}$-$PEG_{2000}$-$NH_2$ (10 mg, 4.41 µmol) and DA (31.5 mg, 250 µmol) were dissolved in 1 mL DMF. Then the pH value was adjusted to ~8.5 by the addition of 1 M NaOH. The solution was stirred overnight in the dark at room temperature under nitrogen. Next, the solution was transferred to a dialysis bag (MWCO 500 Da) and then subjected to dialysis against water at pH 8.5. The product of $C_{18}$-$PEG_{2000}$-DA was obtained by freeze-drying. $^1$H NMR of $C_{18}$-$PEG_{2000}$-DA (400 MHz, $CDCl_3$, Supplementary Fig. 31) δ (ppm): 3.66 (m, 118H), 2.09 (s, 3H), 2.08 (s, 3H), 1.26 (m, 59H), 0.89 (t, $J$ = 6.5 Hz, 3H). The characteristic peaks of double bond at 2918 $cm^{-1}$ and carbonyl group at 1683 $cm^{-1}$ derived from DA moiety of $C_{18}$-$PEG_{2000}$-DA were verified by Fourier transform infrared spectroscopy (FTIR) (Supplementary Fig. 32).

Synthesis of $C_{18}$-$PEG_{2000}$-SA. As a control, $C_{18}$-$PEG_{2000}$-SA was synthesized and purified using the same method. $^1$H NMR of $C_{18}$-$PEG_{2000}$-SA (400 MHz, $CDCl_3$, Supplementary Fig. 33) δ (ppm): 3.66 (m, 118H), 2.60 (br, 2H), 2.54 (br, 2H), 2.19 (t, $J$ = 7.7 Hz, 2H), 1.26 (m, 34H), 0.90 (t, $J$ = 6.5 Hz, 3H). FTIR detected the carbonyl group at 1683 $cm^{-1}$ originated from SA moiety of $C_{18}$-$PEG_{2000}$-SA fragments (Supplementary Fig. 34).

Synthesis of a-RB. RB (100 mg, 0.1 mmol) and 2-ethylhexyl bromide (50 mg, 0.26 mmol) were dissolved in N, N-dimethylformamide and stirred at 80 °C overnight, then concentrated under reduced pressure. The residue was followed by extraction with ethyl acetate and water. The organic layer was dried over $MgSO_4$ and concentrated under reduced pressure. The crude product was further purified by column chromatography with ethyl acetate as eluent to afford the final product a-RB (dark purple powder, 72.0 mg, yield: 66.0%). MALDI-TOF-MS: (m/z): calculated for $C_{28}H_{20}Cl_4I_4NaO_5^+$ $[(M+Na)^+]$: 1108.6107; found 1108.6862. $^1$H NMR (400 MHz, DMSO-$d_6$, Supplementary Fig. 35) δ (ppm): 0.56–0.59 (t, $J$ = 4 Hz, 3H), 0.76–0.80 (t, $J$ = 5.3 Hz, 3H), 0.84–0.93 (m, 4H), 0.97–1.02 (m, 2H), 1.05–1.11 (m, 2H), 1.23 (s, 1H), 3.87–3.88 (d, $J$ = 4 Hz, 2H), 7.47–7.48 (d, $J$ = 4 Hz, 2H).

## Cell imaging

HepG2 cells were incubated with ALCNs or C-ALCNs for 24 h. Then the incubation medium was removed, washed with PBS buffer for three times. Next, 1 × HEPES was added to adjust pH value to 5.5, 6.5, and 7.4, respectively. After 12 h, the cells were collected with different numbers after washing, trypsin digestion and counting. The HepG2 cells were centrifuged at 74 g for 3 min. After aspirating the supernatant, the cells were pre-irradiated with the 808 nm laser (1 W/cm$^2$) or white light (0.1 W/cm$^2$) for 1 min. Afterglow and fluorescence luminescence images were acquired by an IVIS Spectrum imaging system. Afterglow images were captured with a 10 s acquisition time with an open filter and fluorescence images were captured with a 5 s acquisition time with an excitation wavelength at 480 nm and an emission wavelength at 580 nm. The afterglow and fluorescence intensities were quantified by measuring the signal intensity of the region of interest (ROI) using Living Imaging 4.5 Software. The process of the afterglow imaging of 4T1 cells was the same as that of HepG2 cells.

## Animals and tumor models

All animals were purchased from Chang Zhou Cavensla Experimental Animal Technology Co. Ltd. The mice were housed under standard conditions (25 ± 3 °C, 60 ± 10% relative humidity) with 12 h light/dark cycle. The tumor weight should not exceed 10% of normal body weight and the tumor length and width were measured with caliper and tumor volumes were calculated by (length×width$^2$)/2. In all experiments, the tumor volumes did not exceed the maximal tumor size/burden permitted by the Animal Ethics Committee of the Soochow University Laboratory Animal Center.

To establish 4T1 tumors in 8-week-old BALB/c female mice (body weight of 14–16 g), 4T1 cells ($2$–$3 \times 10^6$ cells per mouse) were suspended in 1 × PBS (50 µL). Each mouse was injected subcutaneously on the right lower extremity. Tumors were allowed to grow to about 70 mm$^3$ (approximately 7 days) before experiments. In addition to the different types of mice, the establishment of HepG2-xenograft subcutaneous tumors and B16F10-xenograft subcutaneous tumors were the same as 4T1 tumors, which were performed in 8-week-old BALB/c-Nu female mice and 8-week-old C57BL/6 female mice, respectively.

To establish orthotopic liver model in 8-week-old BALB/c-Nu male mice (body weight of 16–20 g), a midline incision of the anterior abdominal wall was made. HepG2 cells ($6$ - $8 \times 10^5$ cells per mouse, suspended in 25 µL 1 × PBS) were carefully injected into the lube of liver of mice under anesthesia by isoflurane. Tumor growths were monitored by MRI imaging. After 7 days, mice with orthotopic liver model were successfully established for afterglow and fluorescence imaging.

Pulmonary metastasis models were established by tail vein injecting 4T1$^{Luc}$ cells ($1$ - $2 \times 10^5$ cells, suspended in 200 µL 1× PBS) in 4 - 6-week-old BALB/c female mice (body weight of 14–16 g). Tumor growths were traced by D-luciferin. After 7 days, mice with pulmonary metastasis models were successfully used for afterglow and fluorescence imaging. Noted that the study was independent of sex or gender.

## In vivo imaging

Test sizes were three mice per treatment, balancing sufficient replication of results with a reduction in mice number. All mice images were included in the analyses. Cages of tumor-bearing mice were randomly selected for the following treatments. ALCNs or C-ALCNs (1 mg/kg ASN or C-ASN, and 0.1 mg/kg AIN) was injected through intra-tumor and intra-muscle in 4T1-xenograft subcutaneous tumor-bearing female mice or HepG2-xenograft subcutaneous tumors in male mice ($n$ = 3 mice each group). ALCNs or C-ALCNs (4 mg/kg ASN or C-ASN, and 0.4 mg/kg AIN) was systematically injected through the tail vein in orthotopic liver tumors models ($n$ = 3 mice each group). Bio-ALCNs, ALCNs or C-ALCNs (4 mg/kg Bio-ASN, ASN or C-ASN, and 0.4 mg/kg Bio-AIN, AIN) were systematically injected through the tail vein in B16F10-xenograft subcutaneous tumors or pulmonary metastasis models ($n$ = 3 mice each group) in female mice. Afterglow and fluorescence luminescence images were acquired at t = 0, 0.5, 1, 2, 4, 8, 12, 24, and 48 h post-injection. Before acquiring afterglow luminescence images, the mice were illuminated for 1 min with an 808 nm laser at a power density of 0.5 W/cm$^2$ or white light at a power density of 0.1 W/cm$^2$. Afterglow luminescence images were captured with a 30 s acquisition time with an open filter using an IVIS Spectrum imaging system. Fluorescence images were captured with a 5 s acquisition time with an excitation wavelength at 480 nm and emission wavelength at 580 nm. The afterglow and fluorescence luminescence intensities for each individual organ were analyzed by the ROI analysis using the Living Image 4.5 Software.

## Data analysis

Results are expressed as the mean ± s.d. unless otherwise stated. For comparisons, Student's two-sided $t$ test and one-way ANOVA analysis were performed. For all tests, $P < 0.05$ was considered statistically significant. We collected data from the animal studies in a blinded manner, and no data were excluded from the final statistical analysis. All statistical calculations were performed using Origin, including assumptions of tests used (OriginLab Corporation, USA).

## Reporting summary

Further information on research design is available in the Nature Portfolio Reporting Summary linked to this article.

## Data availability

The data supporting the findings of this study are available within the paper and Supplementary Information files. Any raw data files in another format are available from the corresponding author. Source data are provided with this paper.

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

## Acknowledgements

This work was supported by the financial support from National Natural Science Foundation of China (22274107 from Q.M. and 22106115 from Q.L.), China Postdoctoral Science Foundation (2021TQ0234, L.Y.), Jiangsu Specially Appointed Professorship (Q.M.), Outstanding Youth Fund of Jiangsu Province (BK20230009, Q.M.), Natural Science Foundation of Jiangsu Province (BK20190811 from Q.M. and BK20190830 from Q.L.), Leading Talents of Innovation and Entrepreneur-ship of Gusu (ZXL2021457, Q.M.) and Soochow Technological Project (SYS2020082, Q.L.).

## Author contributions

Q.M. conceived and supervised the project. Y.J., M.Z. and J.M. carried out experiments. Y.J., M.Z., J.M., W.C., Y.Z., M.M., Y.L., Q.L. and Q.M. analyzed and interpreted the data. Q.M. wrote the manuscript.

## Competing interests

The authors declare no competing interests.
