## [Peer Review File · Nature Communications]

Reviewers' Comments:

Reviewer #1:

Remarks to the Author:

Jiang et al. developed ALCNs with acidity-activatable upconversion afterglow luminescence for tumor imaging. ALCNs use pH-controlled proximity of donor and acceptor nanoparticles to achieve singlet oxygen transfer from initiator to substrate within the nanoparticles upon 808 nm laser pre-irradiation. This work is an interesting extension of previously published work by the authors (J. Am. Chem. Soc. 2022, 144, 6719–6726). In my opinion, this work has high novelty and will attract great interest from the readers. However, I have several questions and concerns about this manuscript.

Comments:

1. The authors should elucidate the specific mechanism of pH-mediated afterglow luminescence. Please elaborate on the "off-on" signals of ALCNs, particularly "off".
2. Is the acid-activatable afterglow luminescence of ALCNs reversible? For instance, after ALCNs are pre-irradiated and have achieved afterglow luminescence in acidic conditions, can the afterglow be "turned off" after returning to neutral pH? In addition, can ALCNs be pre-irradiated *ex vivo* before systemic administration and be activated upon reaching the acidic tumor microenvironment?
3. Please indicate the deuterated reagent used for ¹H-NMR.
4. Please provide the FTIR spectra of C18-PEG2000-DA and C18-PEG2000-SA.
5. Does the concentration of ALCNs, NIR power intensity, and laser duration influence the afterglow intensity and/or duration? It would be better to demonstrate more trials under different conditions (e.g. different concentrations, laser power, irradiation time, etc.)
6. The pH inside cancer cells is generally acidic, the authors should justify why the pH is adjusted during *in vitro* experiments. If the author feels that the afterglow luminescence effect is not obvious when the pH is not adjusted, please also include the *in vitro* luminescence intensity of the cancer cell without pH adjustment as the control.
7. It would be beneficial to test ALCNs on other cancer types other than breast cancer for broad-spectrum usage.

Reviewer #2:

Remarks to the Author:

In the article entitled "Acidity-activatable upconversion afterglow luminescence cocktail nanoparticles for ultrasensitive *in vivo* imaging", organic afterglow luminescence cocktail nanoparticles (ALCNs) for pH-activatable are reported. ALCNs comprised afterglow initiator nanoparticles (AIN) and afterglow substrate nanoparticles (ASN), where the acid-cleavable bond on ASN was used to control the proximity of ASN to AIN through electrostatic interaction for ¹O₂ transfer between the nanoparticles and activation of subsequent afterglow luminescence. Therefore, ALCNs were applicable for *in vivo* imaging of 4T1-xenograft subcutaneous tumor and orthotopic liver tumors with a high SNR. The idea reported here is novel, which could well expand afterglow imaging for sensitive *in vivo* imaging. This paper is suggested for the publication in Nat. Commun. with minor revision.

1. To verify that proximity of ASN to AIN brings about ¹O₂ transfer, it would be better to display the size changes of ALCNs during pH-responsive process.
2. The energy transfer is a kind of nonradiative process, which does not occur through photon transfer. Please check on the FRET principle and describe the energy transfer in a correct definition.
3. Please add the excitation condition in the caption for Fig 2e.
4. To avoid misunderstanding, afterglow spectra of ALCNs at different pH values should not be normalized (Fig 2f).
5. Where are the ALCNs located after internalization into the cells? And after that, would AIN and ASN be colocalized? Some fluorescence imaging studies are suggested.
6. Why were the 4T1 cells, different from the cells used for cell imaging, adopted for subcutaneous tumor model?
7. For *in vivo* imaging of 4T1-xenograft tumors, ALCNs were intratumorally but not intravenously injected. Is it because that AIN and ASN could hardly be kept together during blood circulation? Some

discussion on this point would be helpful.

8. What does the dash line in the H&E staining images mean? The images display just the pure tumor slices or a hybrid of tumor and liver tissue?

9. It would be better to provide fluorescent images of tumor tissues, which are used to determine where the ALCNs locate and whether AIN and ASN are colocalized for efficient $^{1}O_2$ transfer.

Reviewer #3:

Remarks to the Author:

Jiang et al. presented a new nanoparticle cocktail for afterglow luminescence imaging. Within the cocktail, two nanoparticles carrying the singlet oxygen donor and acceptor respectively are separated by electrostatic repulsion. Only after the acceptor particle is activated by low pH and converted into positively charged, the two particles would come into proximity and promote singlet oxygen transfer. Though afterglow luminescence imaging does have some promising applications, the idea of using a nanoparticle cocktail is a little too complicated and the benefits are not that obvious. The authors failed to address some of the potential limitations and pitfalls of their system. The major claims of this paper lack evidence support.

The authors claim the biggest benefit of this cocktail compared to using a single nanoparticle is the transfer of singlet oxygen is more efficient than the transfer of electrons to quenchers over longer distances. The authors cited two references claiming that singlet oxygen can be transferred up to 200 nm. However, these references do not use singlet oxygen transfer in vivo. When the particles are injected in vivo, there is a mere chance that the singlet oxygen can be transferred over 200 nm considering the complex mixture of other molecules that are around and the short half-life of singlet oxygen. I would expect the efficiency of this singlet oxygen transfer between two nanoparticles to be low. Thus, the benefits the authors claimed are unconvincing.

The two particles have a big size disparity: one is 100 nm and the other is 20 nm, therefore, the biodistribution and retention of the two particles are expected to be different. In order for the cocktail system to work efficiently, the two particles would need to be in proximity to each other. If the particles have different biodistributions, the probability or efficiency for these two types of particles to come together and stay together for a period of time would be extremely low in vivo.

The authors claim their cocktail system can significantly improve the SNR of imaging due to the more efficient transfer of singlet oxygen. The SNR in the animal imaging did not seem to improve significantly compared to the previously reported single nanoparticle system.

Therefore this work is not suitable for this journal due to these major issues.

Point-by-point Response to Reviewers' Comments (NCOMMS-23-37148)

Reviewer #1

Comments:

Jiang *et al.* developed ALCNs with acidity-activatable upconversion afterglow luminescence for tumor imaging. ALCNs use pH-controlled proximity of donor and acceptor nanoparticles to achieve singlet oxygen transfer from initiator to substrate within the nanoparticles upon 808 nm laser pre-irradiation. This work is an interesting extension of previously published work by the authors (*J. Am. Chem. Soc.* 2022, 144, 6719–6726). In my opinion, this work has high novelty and will attract great interest from the readers. However, I have several questions and concerns about this manuscript.

Response: We really appreciate and agree with the reviewer's constructive comments.

1. The authors should elucidate the specific mechanism of pH-mediated afterglow luminescence. Please elaborate on the "off-on" signals of ALCNs, particularly "off".

Response: Thanks for the reviewer's constructive comments. In Figure 1, we illustrated the specific mechanism of pH-mediated afterglow luminescence based on the transfer of singlet oxygen inside ALCNs, especially when the afterglow signal of ALCNs is off.

Revised Figure 1.

Page 2, left panel, line 10: Under physiological conditions, both the donor AIN and acceptor ASN in ALCNs are negatively charged, which induces the separation of AIN and ASN via

electrostatic repulsion. This blocks $^1\text{O}_2$ transfer from the AIN to ASN due to the limited transfer distance of $^1\text{O}_2$ in aqueous solution, preventing the downstream oxidation of afterglow substrate to form a dioxetane intermediate and generation of afterglow luminescence.

2. Is the acid-activatable afterglow luminescence of ALCNs reversible? For instance, after ALCNs are pre-irradiated and have achieved afterglow luminescence in acidic conditions, can the afterglow be “turned off” after returning to neutral pH? In addition, can ALCNs be pre-irradiated *ex vivo* before systemic administration and be activated upon reaching the acidic tumor microenvironment?

Response: We appreciate the reviewer’s comments. Actually, ALCNs is irreversible because the acid-labile amide bond of ASN is hydrolyzed to liberate the amino-termini group, which is irreversible to induce the surface charge changes from negative to positive under the acidic environment (*Nat. Commun.* 2022, 13, 2038; *Adv. Mater.* 2018, 30, 1705436; *Acc. Chem. Res.* 2018, 51, 2848-2856; *ACS Nano*, 2018, 12, 12401-12415). However, if some groups such as pH-sensitive tertiary amines are used as the charge-reversible group (*Nat. Nanotechnol.* 2017, 12, 648-654; *Nat. Commun.* 2020, 11, 5828; *Angew. Chem. Int. Ed.* 2022, 61, e202200152), the reversibility of acid-activatable afterglow luminescence can be achieved. For the pre-irradiation for *in vivo* imaging, in the off state of ALCNs, the light-pre-irradiated AIN-produced $^1\text{O}_2$ could not be transferred to ASN to form the dioxetane intermediate, thus no activating signal upon reaching the acidic tumor microenvironment. We will construct new systems for pH-reversible activatable afterglow luminescence and pre-irradiated system in the further study.

3. Please indicate the deuterated reagent used for ^1H -NMR.

Response: We appreciate the reviewer’s suggestion. We used deuterated chloroform as a deuterated reagent and have added the relevant information to Supplementary Fig. 24 and 26.

Supplementary Fig. 24. ^1H NMR spectrum of C_{18} -PEG $_{2000}$ -DA in CDCl_3 .

Supplementary Fig. 26. ^1H NMR spectrum of $\text{C}_{18}\text{-PEG}_{2000}\text{-SA}$ in CDCl_3 .

4. Please provide the FTIR spectra of $\text{C}_{18}\text{-PEG}_{2000}\text{-DA}$ and $\text{C}_{18}\text{-PEG}_{2000}\text{-SA}$.

Response: We appreciate the reviewer's suggestion. We have added the FTIR spectrum of $\text{C}_{18}\text{-PEG}_{2000}\text{-DA}$ and $\text{C}_{18}\text{-PEG}_{2000}\text{-SA}$ in the revised manuscript.

Page 9, left panel, line 23: The characteristic peaks of double bond at 2918 cm^{-1} and carbonyl group at 1683 cm^{-1} derived from DA moiety of $\text{C}_{18}\text{-PEG}_{2000}\text{-DA}$ were verified by Fourier transform infrared spectroscopy (FTIR) (Supplementary Fig. 25).

Page 9, left panel, line 28: FTIR detected the carbonyl group at 1683 cm^{-1} originated from SA moiety of $\text{C}_{18}\text{-PEG}_{2000}\text{-SA}$ fragments (Supplementary Fig. 27).

Supplementary Fig. 25. The FTIR spectrum of $\text{C}_{18}\text{-PEG}_{2000}\text{-DA}$ mixed in a potassium bromide tablet.

Supplementary Fig. 27. The FTIR spectrum of $\text{C}_{18}\text{-PEG}_{2000}\text{-SA}$ mixed in a potassium bromide tablet.

5. Does the concentration of ALCNs, NIR power intensity, and laser duration influence the afterglow intensity and/or duration? It would be better to demonstrate more trials under different conditions (e.g. different concentrations, laser power, irradiation time, etc.)

Response: Thanks for the reviewer's sincere comments. According to the reviewer's suggestion, except for the ratio of ASN and AIN within the ALCNs, we have added the optimized conditions including the laser irradiation time, power density as well as the performance of concentration-dependent afterglow signal. We revised the manuscript accordingly.

Page 4, right panel, line 17 from the bottom: In addition, the ratio of ASN and AIN within the ALCNs, laser irradiation time and power density were optimized and a concentration-dependent afterglow signal was observed (Supplementary Fig. 9).

Supplementary Fig. 9. The afterglow intensity of ALCNs under different conditions. **a** The concentration ratio of ASN and AIN corresponds to the afterglow intensities under 808 nm laser pre-irradiation in $1 \times$ HEPES buffer. **b** Fitted calibration curve of the afterglow intensity of ALCNs after irradiation by 808 nm (1 W/cm^2) laser as a function of the concentration in $1 \times$ HEPES buffer. **c** Afterglow luminescence of ALCNs ($20 \mu\text{g/mL}$ ASN and $2 \mu\text{g/mL}$ AIN) after illumination by 808 nm laser at different power densities in $1 \times$ HEPES buffer. **d** Afterglow luminescence of ALCNs ($20 \mu\text{g/mL}$ ASN and $2 \mu\text{g/mL}$ AIN) at different light irradiation time by 808 nm laser (1 W/cm^2) in $1 \times$ HEPES buffer. The error bars represent the standard deviation ($n = 3$).

6. The pH inside cancer cells is generally acidic, the authors should justify why the pH is adjusted during in vitro experiments. If the author feels that the afterglow luminescence effect is not obvious when the pH is not adjusted, please also include the in vitro luminescence intensity of the cancer cell without pH adjustment as the control.

Response: Thanks for the reviewer's comments. To mimic the extracellularly acidic microenvironment, we choose to adjust pH in the experiment. According to the reviewer's suggestion, we have added the group without pH adjustment in revised Figure 3, which

presented similar result to that of group with regulated pH 7.4. This may be caused by its relatively low cellular uptake of ASN and AIN within cells.

Page 5, left panel, Line 10: No obviously activated afterglow luminescence in the groups after incubation with medium and 7.4 was observed, indicating extracellularly acidic microenvironment indeed facilitates the charge converse and nanoparticle proximity for activated afterglow signal in tumor cells.

Revised Figure 3.

7. It would be beneficial to test ALCNs on other cancer types other than breast cancer for broad-spectrum usage.

Response: Thanks for the reviewer's comments. Except for breast tumor imaging, we further explored and demonstrated the feasibility of ALCNs for tumor imaging in HepG2-tumor-bearing mice. We revised the manuscript accordingly.

Page 5, right panel, line 6: Remarkably, for both HepG2 and 4T1 cells, the activated afterglow luminescence intensities of ALCNs could be linearly correlated with the cell numbers (Fig. 3f and Supplementary Fig. 11-13).

Page 8, left panel, line 27: The acidic microenvironment-activatable upconversion afterglow luminescence was also observed in HepG2-tumor-bearing mice, verifying the feasibility and broad application of ALCNs for tumor imaging (Supplementary Fig. 18, 19).

Supplementary Fig. 12. In vitro afterglow luminescence imaging capability of ALCNs towards 4T1 cells. **a** Afterglow luminescence images of cell pellets after incubation with ALCNs or C-ALCNs (30 μ g/mL ASN or C-ASN, and 3 μ g/mL AIN) at different pH values and without pH adjustment under 808 nm pre-irradiation (upper panel) or white light pre-irradiation (middle panel), and corresponding fluorescence images (bottom panel). The fluorescence images were acquired with excitation at 480 nm and emission wavelength at 580 nm. **b-d** Quantification of afterglow intensities under 808 nm laser pre-irradiation (**b**), under white light pre-irradiation (**c**), fluorescence intensities (**d**) of 4T1 cells incubation of ALCNs and C-ALCNs. **e** Linear fitting curve of afterglow intensities with different cell numbers under 808 nm laser pre-irradiation of ALCNs in 1 \times HEPES buffer (pH = 5.5). Statistical analysis was performed with one-way ANOVA ($n = 3$, ns: not statistically significant, ** $p < 0.01$, *** $p < 0.001$). The error bars represent the standard deviation.

Supplementary Fig. 13. Afterglow imaging of 4T1 cells. Afterglow images under 808 nm laser pre-irradiation of ALCNs and C-ALCNs (30 μ g/mL ASN or C-ASN, and 3 μ g/mL AIN) incubated with different numbers of 4T1 cells in 1 \times HEPES buffer (pH = 5.5).

Supplementary Fig. 18. In vivo afterglow luminescence imaging of HepG2-xenograft tumors. a Afterglow images under an 808 nm pre-irradiation (left panel), white light pre-irradiation (middle panel), and fluorescence images (right panel) after intratumor and intramuscular injection of ALCNs or C-ALCNs (1 mg/kg ASN or C-ASN, and 0.1 mg/kg AIN). The fluorescence images were acquired with an excitation wavelength at 480 nm and emission wavelength at 580 nm. **b** Ex vivo afterglow images of various tissues under an 808 nm and white light pre-irradiation from HepG2-xenograft subcutaneous tumor-bearing mice at 48 h post-injection of ALCNs or C-ALCNs. **c-e** The quantified SNRs for afterglow luminescence imaging under an 808 nm pre-irradiation (c), white light pre-irradiation (d), fluorescence luminescence imaging (e) of HepG2-xenograft subcutaneous tumor-bearing mice as a function of time. Statistical analysis was performed with two-way ANOVA ($n = 3$, ns: not statistically significant, $***p < 0.001$). **f** Quantification of ex vivo afterglow intensities of various tissues in Supplementary Fig. 18a. Statistical analysis was performed with one-way ANOVA ($n = 3$, ns: not statistically significant, $***p < 0.001$). The error bars represent the standard deviation.

Supplementary Fig. 19. Study of biodistribution. a Ex vivo fluorescence images of various organs and tissues from an HepG2-xenograft subcutaneous tumor-bearing mouse intratumor injection of ALCNs and C-ALCNs (1 mg/kg ASN or C-ASN, and 0.1 mg/kg AIN) at 48 h with an excitation at 480 nm and emission at 580 nm. **b** Quantification of fluorescence intensities of organs and tissues. The error bars represent the standard deviation ($n = 3$, ns: not statistically significant).

Reviewer #2

Comments:

In the article entitled “Acidity-activatable upconversion afterglow luminescence cocktail nanoparticles for ultrasensitive in vivo imaging”, organic afterglow luminescence cocktail nanoparticles (ALCNs) for pH-activatable are reported. ALCNs comprised afterglow initiator nanoparticles (AIN) and afterglow substrate nanoparticle (ASN), where the acid-cleavable bond on ASN was used to control the proximity of ASN to AIN through electrostatic interaction for $^1\text{O}_2$ transfer between the nanoparticles and activation of subsequent afterglow luminescence. Therefore, ALCNs were applicable for in vivo imaging of 4T1-xenograft subcutaneous tumor and orthotopic liver tumors with a high SNR. The idea reported here is novel, which could well expand afterglow imaging for sensitive in vivo imaging. This paper is suggested for the publication in Nat. Commun. with minor revision.

Response: We really appreciate and agree with the reviewer’s constructive comments.

1. *To verify that proximity of ASN to AIN brings about $^1\text{O}_2$ transfer, it would be better to display the size changes of ALCNs during pH-responsive process.*

Response: Thanks for the reviewer’s comments. We used dynamic light scattering (DLS) to characterize the size changes of ALCNs in acidic conditions. The corresponding results have been added to the manuscript.

Page 4, right panel, line 1: Meantime, an obvious size increase of ALCNs was observed with decreasing the pH values, further demonstrating the electrostatic attraction-mediated proximity and aggregation of ASN and AIN in acidic conditions (Supplementary Fig. 7).

Supplementary Fig. 7. The size changes of ALCNs with different concentrations during pH-responsive process. a-c The hydrodynamic sizes of ALCNs (5 μg/mL ASN and 0.5 μg/mL AIN) (a), ALCNs (10 μg/mL ASN and 1 μg/mL AIN) (b), and ALCNs (20 μg/mL ASN and 2 μg/mL AIN) (c) at different pH values in 1 × HEPES buffer.

2. *The energy transfer is a kind of nonradiative process, which does not occur through photon transfer. Please check on the FRET principle and describe the energy transfer in a correct definition.*

Response: Thanks for the reviewer's comments. We have rephrased the sentence in the revised manuscript.

Page 1, right panel, line 1 from the bottom: Therefore, a novel activatable afterglow approach that breaks such harsh distance limitation of energy transfer is urgently needed to expand versatility and improve sensing capability.

Page 2, left panel, line 2 from the bottom: Beyond conventional energy transfer with a distance requisition within 10 nm between a donor and an acceptor, $^1\text{O}_2$ can be delivered within 200 nm in aqueous solution^{57,58} and 10~55 nm in living cells⁵⁹⁻⁶¹, making feasibility and flexibility of the system for imaging usage.

Page 8, right panel, line 14 from the bottom: ALCNs utilized acidic pH-controlled $^1\text{O}_2$ transfer from donor to acceptor with a longer required distance of 200 nm relative to conventional energy transfer necessitating a distance between a donor and acceptor within 10 nm, which was more feasible and efficient.

3. Please add the excitation condition in the caption for Fig 2e.

Response: Thanks for the reviewer's comments. We have added the information in the figure caption.

Page xx line xx (the caption of Fig. 2e): Fluorescence spectra of ALCNs (10 $\mu\text{g}/\text{mL}$ ASN, and 1 $\mu\text{g}/\text{mL}$ AIN) at different pH values in 1 \times HEPES buffer with an excitation wavelength at 480 nm.

4. To avoid misunderstanding, afterglow spectra of ALCNs at different pH values should not be normalized (Fig 2f).

Response: Thanks for the reviewer's comments. We revised Figure 2f accordingly in the revised manuscript.

Fig. 2f | Afterglow spectra of ALCNs at different pH values in 1 × HEPES buffer.

5. Where are the ALCNs located after internalization into the cells? And after that, would AIN and ASN be colocalized? Some fluorescence imaging studies are suggested.

Response: Thanks for the reviewer's comments. The colocalization study has been performed. As shown in Supplementary Fig. 15, ASN and AIN mainly co-localized in lysosomes. We revised the manuscript accordingly.

Page 5, right panel, line 9: As the emission of NCBS at 775 nm can't be detected by the confocal microscope, R-AIN was prepared with doped alkylated rose bengal (a-RB) to monitor its subcellular colocalization (Supplementary Fig. 14). Co-localization study showed that ASN and R-AIN colocalized mainly in lysosomes (Supplementary Fig. 15).

Supplementary Fig. 15. Subcellular localization study of ALCNs. Confocal fluorescence images of HepG2 cells treated with ASN (40 $\mu\text{g}/\text{mL}$), R-AIN (40 $\mu\text{g}/\text{mL}$) and Hoechst 33342 (10 μM), which were co-stained with commercial Lyso Green Tracker (10 μM) (a), Mito Green Tracker (10 μM) (b), respectively. For Lyso Green Tracker and Mito Green Tracker, an excitation wavelength was 488 nm with emission wavelengths at 510 nm \pm 10 nm. For ASN, an excitation wavelength was 488 nm with emission wavelengths at 580 nm \pm 10 nm. For R-AIN, an excitation wavelength was 559 nm with emission wavelengths at 580 nm \pm 10 nm. Line-scan analyses of merged fluorescence images in (a) and (b) were performed by Image J. The scale bar represents 10 μm .

6. *Why were the 4T1 cells, different from the cells used for cell imaging, adopted for subcutaneous tumor model.*

Response: Thanks for the reviewer's comments. To be consistency and further demonstrate the feasibility of ALCNs for diverse tumors, we have supplemented cell imaging using 4T1 cells and subcutaneous tumor imaging in HepG2-tumor-bearing mice. We revised the manuscript accordingly.

Page 5, right panel, line 6: Remarkably, for both HepG2 and 4T1 cells, the activated afterglow luminescence intensities of ALCNs could be linearly correlated with the cell numbers (Fig. 3f and Supplementary Fig. 11-13).

Page 8, left panel, line 27: The acidic microenvironment-activatable upconversion afterglow luminescence was also observed in HepG2-tumor-bearing mice, verifying the feasibility and broad application of ALCNs for tumor imaging (Supplementary Fig. 18, 19).

Supplementary Fig. 12. In vitro afterglow luminescence imaging capability of ALCNs towards 4T1 cells. **a** Afterglow luminescence images of cell pellets after incubation with ALCNs or C-ALCNs (30 μ g/mL ASN or C-ASN, and 3 μ g/mL AIN) at different pH values and without pH adjustment under 808 nm pre-irradiation (upper panel) or white light pre-irradiation (middle panel), and corresponding fluorescence images (bottom panel). The fluorescence images were acquired with excitation at 480 nm and emission wavelength at 580 nm. **b-d** Quantification of afterglow intensities under 808 nm laser pre-irradiation (**b**), under white light pre-irradiation (**c**), fluorescence intensities (**d**) of 4T1 cells incubation of ALCNs and C-ALCNs. **e** Linear fitting curve of afterglow intensities with different cell numbers under 808 nm laser pre-irradiation of ALCNs in 1 \times HEPES buffer (pH = 5.5). Statistical analysis was performed with one-way ANOVA ($n = 3$, ns: not statistically significant, ** $p < 0.01$, *** $p < 0.001$). The error bars represent the standard deviation.

Supplementary Fig. 13. Afterglow imaging of 4T1 cells. Afterglow images under 808 nm laser pre-irradiation of ALCNs and C-ALCNs (30 μ g/mL ASN or C-ASN, and 3 μ g/mL AIN) incubated with different numbers of 4T1 cells in 1 \times HEPES buffer (pH = 5.5).

Supplementary Fig. 18. In vivo afterglow luminescence imaging of HepG2-xenograft tumors. a Afterglow images under an 808 nm pre-irradiation (left panel), white light pre-irradiation (middle panel), and fluorescence images (right panel) after intratumor and intramuscular injection of ALCNs or C-ALCNs (1 mg/kg ASN or C-ASN, and 0.1 mg/kg AIN). The fluorescence images were acquired with an excitation wavelength at 480 nm and emission wavelength at 580 nm. **b** Ex vivo afterglow images of various tissues under an 808 nm and white light pre-irradiation from HepG2-xenograft subcutaneous tumor-bearing mice at 48 h post-injection of ALCNs or C-ALCNs. **c-e** The quantified SNRs for afterglow luminescence imaging under an 808 nm pre-irradiation (c), white light pre-irradiation (d), fluorescence luminescence imaging (e) of HepG2-xenograft subcutaneous tumor-bearing mice as a function of time. Statistical analysis was performed with two-way ANOVA ($n = 3$, ns: not statistically significant, $***p < 0.001$). **f** Quantification of ex vivo afterglow intensities of various tissues in Supplementary Fig. 18a. Statistical analysis was performed with one-way ANOVA ($n = 3$, ns: not statistically significant, $***p < 0.001$). The error bars represent the standard deviation.

Supplementary Fig. 19. Study of biodistribution. a Ex vivo fluorescence images of various organs and tissues from an HepG2-xenograft subcutaneous tumor-bearing mouse intratumor injection of ALCNs and C-ALCNs (1 mg/kg ASN or C-ASN, and 0.1 mg/kg AIN) at 48 h with an excitation at 480 nm and emission at 580 nm. **b** Quantification of fluorescence intensities of organs and tissues. The error bars represent the standard deviation ($n = 3$, ns: not statistically significant).

7. For *in vivo* imaging of 4T1-xenograft tumors, ALCNs were intratumorally but not intravenously injected. Is it because that AIN and ASN could hardly kept together during blood circulation? Some discussion on this point would be helpful.

Response: We appreciate the reviewer's comments. For the 4T1-xenograft subcutaneous tumor model, intratumor injection was adopted because it can effectively gather the donor and acceptor nanoparticles for acidic response as compared with systemic administration (Figure 4). In the second orthotopic liver tumor model, systemic injection of ALCNs was also used for tumor imaging. Even though a slight difference of plasma half-life of AIN and ASN, a high tumor targeting efficiency and pH-activatable afterglow luminescence was achieved with high imaging SNR (Figure 5).

Page 6, right panel, line 2: Intratumor injection was adopted as it can effectively gather the donor and acceptor nanoparticles for acidic response as compared with systemic administration.

8. What does the dash line in the H&E staining images mean? The images display just the pure tumor slices or a hybrid of tumor and liver tissue?

Response: We appreciate the reviewer's comments. The images showed a hybrid of normal liver tissue and tumor and the dash line means the boundary of them. We provided an explanation and revised the figure accordingly.

Fig. 5h | H&E staining images of tumor slices of mice after injection with ALCNs or C-ALCNs (N: normal tissue, T: tumor tissue). The scale bar represents 100 μm . The error bars represent the standard deviation.

9. *It would be better to provide fluorescent images of tumor tissues, which are used to determine where the ALCNs locate and whether AIN and ASN are colocalized for efficient $^1\text{O}_2$ transfer.*

Response: We appreciate the reviewer's constructive comments. The fluorescent images of tumors slices were shown below, ASN (green) and AIN (red) exhibited good co-localization in TME.

Page 8, left panel, line 21: Moreover, the fluorescence images of tumor sections demonstrated obvious co-localization of ASN and AIN both intracellularly and extracellularly, implying the high possibility of nanoparticle proximity for $^1\text{O}_2$ transfer and activatable afterglow luminescence of ALCNs (Supplementary Fig. 17).

Page 8, right panel, line 28 from the bottom: Clear co-localization of ASN and AIN was also seen in the fluorescence images of tumor section, revealing the high tumor targeting and high potentiality of nanoparticle proximity for afterglow activation after systemic administration (Supplementary Fig. 17).

Supplementary Fig. 17. Fluorescence imaging of tumor sections. Fluorescent images of 4T1 subcutaneous tumor and HepG2 orthotopic tumor sections after treatment with ALCNs by Digital pathology scanner. ASN (green, 1 mg/kg for subcutaneous tumor, and 4 mg/kg for orthotopic tumor, with an excitation at 490 nm and emission at 570 ± 10 nm), AIN (red, 0.1 mg/kg for subcutaneous tumor, and 0.4 mg/kg for orthotopic tumor, with an excitation at 750 nm and emission at 780 ± 10 nm), nucleus (blue, 10 μ M, with an excitation at 350 nm and emission at 420 ± 10 nm). The scale bar represents 5 μ m. Dashed and solid arrows indicate the intracellular and extracellular co-localization, respectively.

Reviewer #3

Comments:

Jiang et al. presented a new nanoparticle cocktail for afterglow luminescence imaging. Within the cocktail, two nanoparticles carrying the singlet oxygen donor and acceptor respectively are separated by electrostatic repulsion. Only after the acceptor particle is activated by low pH and converted into positively charged, the two particles would come into proximity and promote singlet oxygen transfer. Though afterglow luminescence imaging does have some promising applications, the idea of using a nanoparticle cocktail is a little too complicated and the benefits are not that obvious. The authors failed to address some of the potential limitations and pitfalls of their system. The major claims of this paper lack evidence support.

Response: We really appreciate the opinions of the reviewers for bringing us in-depth brainstorming. This study proposes an effective and general platform for designing activatable afterglow luminescence probes. As compared with fluorescence imaging and non-activatable afterglow probes, ALCNs enable superior imaging performance with high sensitivity and specificity, holding great potential for precise biomedical imaging.

The authors claim the biggest benefit of this cocktail compared to using a single nanoparticle is the transfer of singlet oxygen is more efficient than the transfer of electrons to quenchers over longer distances. The authors cited two references claiming that singlet oxygen can be transferred up to 200 nm. However, these references do not use singlet oxygen transfer in vivo. When the particles are injected in vivo, there is a mere chance that

the singlet oxygen can be transferred over 200 nm considering the complex mixture of other molecules that are around and the short half-life of singlet oxygen. I would expect the efficiency of this singlet oxygen transfer between two nanoparticles to be low. Thus, the benefits the authors claimed are unconvincing.

Response: We appreciate the reviewer's comments. It has been widely reported that $^1\text{O}_2$ can diffuse ~200 nm in aqueous solution (*J. Am. Chem. Soc.* 1983, 105, 4129-4135. *Nat. Commun.* 2014, 5, 3591; *Photochem. Photobiol.* 1981, 33, 627-634). The transfer distance of $^1\text{O}_2$ in living cells was also reported to be 10~55 nm in literatures (*Adv. Mater.* 2021, 33, 2103627; *J. Biophotonics* 2017, 10, 338; *Phys. Med. Biol.* 2005, 50, 2597–2616), which is still longer than that of electron transfer. We agree with that the shortened transfer distance of $^1\text{O}_2$ induces the decreased efficiency of $^1\text{O}_2$ transfer from donor to acceptor and thus decreased afterglow luminescence. However, based on the tumor imaging in Figure 4 and 5, this strategy is indeed effectual for in vivo utility. We have rephrased the sentence in the revised manuscript.

Page 2, left panel, line 2 from the bottom: Beyond conventional energy transfer with a distance requisition within 10 nm between a donor and an acceptor, $^1\text{O}_2$ can be delivered within 200 nm in aqueous solution^{57,58} and 10~55 nm in living cells⁵⁹⁻⁶¹, making feasibility and flexibility of the system for imaging usage.

2. The two particles have a big size disparity: one is 100 nm and the other is 20 nm, therefore, the biodistribution and retention of the two particles are expected to be different. In order for the cocktail system to work efficiently, the two particles would need to be in proximity to each other. If the particles have different biodistributions, the probability or efficiency for these two types of particles to come together and stay together for a period of time would be extremely low in vivo.

Response: Thanks for the reviewer's comments. The metabolism behavior of AIN and ASN in vivo were studied by investigating the blood clearance kinetics and organ distribution after systemic administration (Supplementary Fig. 20 and 22). Although their in vivo biological behaviors were not completely consistent, they both showed the highest accumulation capacity in the liver tumor due to the similar hepatobiliary metabolism pathway and enhanced permeability retention (EPR) effect of nanoparticles (Supplementary Fig. 22). Clear co-localization of ASN and AIN was also seen in the fluorescence images of tumor section, further revealing the high tumor targeting and high potentiality of nanoparticle proximity for afterglow activation after systemic administration (Supplementary Fig. 17). Therefore, AIN and ASN were more likely to co-locate and achieve effective singlet oxygen delivery for activatable afterglow luminescence in the liver tumor. We added the data and revised the manuscript accordingly.

Page 8, left panel, line 23 from the bottom: Before in vivo imaging, the pharmacokinetics of AIN and ASN were investigated (Supplementary Fig. 20). By endowing different nanoparticle sizes, ASN and AIN established a different plasma half-life of 11.9 and 33.6 min, respectively.

Page 8, right panel, line 22: The biodistribution study of ASN and AIN after systemic administration at 4 and 24 h post-injection of nanoparticles was also performed. As shown in Supplementary Fig. 22, both of ASN and AIN established the highest accumulation capacity in the liver tumor site owing to the hepatobiliary metabolism pathway and EPR effect of nanoparticles, which further demonstrated the feasibility of ALCNs for orthotopic liver tumor imaging. Clear co-localization of ASN and AIN was also seen in the fluorescence images of tumor section, revealing the high tumor targeting and high potentiality of nanoparticle proximity for afterglow activation after systemic administration (Supplementary Fig. 17).

Supplementary Fig. 20. The time dependent blood clearance kinetics of AIN (0.8 mg/kg, with an excitation at 720 nm and emission at 790 nm) (a), and ASN (8 mg/kg, with an excitation at 480 nm and emission at 580 nm) (b). The data were fitted with a two-compartment model. The error bars represent the standard deviation (n = 3).

Supplementary Fig. 22. Biodistribution of ASN and AIN in various organs of orthotopic liver tumors. **a, c** Fluorescence images of ASN (8 mg/kg, with an excitation at 480 nm and emission at 580 nm) and AIN (0.8 mg/kg, with an excitation at 720 nm and emission at 790 nm) in heart, liver, spleen, lungs, kidney, and tumor at 4 h and 24 h. **b, d** Qualifications of fluorescence intensity in (a) and (c).

Supplementary Fig. 17. Fluorescence imaging of tumor sections. Fluorescent images of 4T1 subcutaneous tumor and HepG2 orthotopic tumor sections after treatment with ALCNs by Digital pathology scanner. ASN (green, 1 mg/kg for subcutaneous tumor, and 4 mg/kg for orthotopic tumor, with an excitation at 490 nm and emission at 570 ± 10 nm), AIN (red, 0.1 mg/kg for subcutaneous tumor, and 0.4 mg/kg for orthotopic tumor, with an excitation at 750 nm and emission at 780 ± 10 nm), nucleus (blue, 10 μ M, with an excitation at 350 nm and emission at 420 ± 10 nm). The scale bar represents 5 μ m. Dashed and solid arrows indicate the intracellular and extracellular co-localization, respectively.

3. *The authors claim their cocktail system can significantly improve the SNR of imaging due to the more efficient transfer of singlet oxygen. The SNR in the animal imaging did not seem to improve significantly compared to the previously reported single nanoparticle system. Therefore this work is not suitable for this journal due to these major issues.*

Response: Thanks for the reviewer's comments. As compared with most of previously reported single nanoparticle system (Table 1), ALCNs endow comparable SNR attributing to the strong afterglow luminescence. Though ALCNs endow a weaker afterglow signal relative to the single nanoparticle system, ALCNs possess a biomarker-mediated contrast imaging and can be used for disease-related in vivo imaging with high specificity.

Page 8, right panel, line 7: As compared with most of previously reported organic afterglow system (Supplementary Table 1), ALCNs endow a comparable in vivo SNR and superior specificity for disease-associated in vivo imaging.

System	λ_{em} (nm)	SNR in mice (administration/dose)	Ref.
PPV and its analogs			

MEHPPV/NCBS/PEG- b -PPG- b -PEG NPs	780	419 (s.c./12.5 µg)	[1]
PPV-TPP/PEG- b -PPG- b -PEG NPs	720	27.6 (i.v./80 µg)	[2]
PPV-PEGL/NCBS NPs	775	4170 (s.c./6.5 µg)	[3]
PFPV/TTMN/PEG- b -PPG- b -PEG NPs	630	—	[4]
Thiophene-based SPs			
PFODBT NPs	690-770	50 (i.t./60 µg)	[5]
NIR-3 NPs	800-820	150 (i.t./50 µg)	[6]
AEE			
AEE-1	< 400	150 (i.v./95.3 µg)	[7]
AEE-2/TPE-TV-CyP/DSPE-PEG NPs	625	461.3 (s.c./15 µg)	[8]
AEE-4/TPE-Ph-DCM/DSPE-PEG NPs	640	—	[9]
AEE-5/(TPE-DPA) ₂ -Py/DSPE-PEG NPs	660	—	[10]
AEE-6	600	93.7 (blood/20 µM)	[11]
AEE-7/MB	700	—	[12]
DO and SO			
DO/NCBS/PFVA/PEG- b -PPG- b -PEG NPs	780	2922 (i.v./95.3 µg)	[13]
SO/PdPc(Obu) ₃ /Eu(TPPO) ₂ (β-NTA) ₃ NPs	613	—	[14]
CUEM/SiPc/carboxylated polystyrene NPs	445	131 (s.c./150 µg)	[15]
CLA			
ADLumin-1	660	2189 (s.c./-)	[16]
C ₈ -CLA/NCBS/TTB/DSPE-PEG NPs	780	18 (i.v./70 µg)	[17]
Porphyrin			
Ppa nanomicelles	760	215.1 (i.v./224 µg)	[18]
Ce4/PEG- b -PPG- b -PEG NPs	660	690 (s.c./1.25 µg)	[19]
Rubrene			
Rubrene/Ir-OTf/PEG- b -PPG- b -PEG NPs	560	4 (i.p./4.2 mg)	[20]
This work			
ALCNs	590	632.3 (i.t./16.8 µg) 224.3 (i.v./67.2 µg)	

Supplementary Table 1. SNR of representative molecular afterglow imaging probes.

Reviewers' Comments:

Reviewer #1:

Remarks to the Author:

The authors have addressed all concerns in the revised manuscript and attempted to clarify the major concern regarding the "off-on" signals of ALCNs. However, there are still several misleading texts in the manuscript that should be clarified for the readers before publication.

1. The authors explained the "off-on" signals of ALCNs. For the "off" signal, both AIN and ASN are negatively charged at pH 7.4, which blocks singlet oxygen transfer and thereby prevents afterglow luminescence generation. For the "on" signal, the charge of ASN will be positive while AIN remains negative, resulting in an electrostatic attraction that allows singlet oxygen transfer for downstream oxidation of afterglow substrate to form a dioxetane intermediate and generate afterglow luminescence. To avoid the misunderstanding of the "off-on" signal, I highly recommend the authors clarify this in the manuscript. For instance, the authors could add: NIR unresponsive "off" and NIR responsive "on", or 1O₂ inactivated "off" and 1O₂ activated "on", or anything that could better clarify this point. This should be stated very clearly in the text so that the readers will not misunderstand that the "off-on" signals can be manually controlled.
2. As the authors stated, the pH-responsive mechanism of ALCNs is irreversible due to the hydrolyzed acid-labile amide bond of ASN that liberates the amino-termini group. The authors should discuss this fact in the discussion and include that the "off-on" signals of ALCNs are influenced by the tumor microenvironment.
3. For question 5, the authors have answered the question regarding the afterglow intensity of ALCNs under different conditions in Supplementary Fig. 9. However, the afterglow duration of ALCNs under different conditions is still missing.

Reviewer #2:

Remarks to the Author:

In the revised manuscript, the authors have well addressed the comments raised. The new data collected have well improved the revised manuscript. The paper is suggested for the accept for the publication at current version.

Reviewer #3:

Remarks to the Author:

In this revised manuscript, the authors added some more data in an attempt to address some of the limitations of their nanoparticle system. Though the added data provided some clarifications, it is still unconvincing that the biggest limitations have been addressed. The two-particle system dictates the largest obstacle for this system is to bring the two particles close enough in vivo for singlet oxygen transfer to happen. It can only happen using specific animal models as the authors showed in their liver tumor model. This nanoparticle system lacks general application unless these limitations can be addressed. Here are my responses to the authors' revision:

1. I appreciate that the authors provided more references stating the diffusion distance of singlet oxygen is around 10-55 nm in living cells. This means the two nanoparticles will have to come into at least this distance for the singlet oxygen transfer to happen. When the nanoparticles are administered systemically, it is very difficult to ensure this would happen.
2. In addition to the previously provided subcutaneous tumor model with intratumoral injection of the nanoparticles, the authors provided an orthotopic liver tumor model with systematic administration of nanoparticles. The orthotopic hepatic tumor model is not a very convincing model to use in this case since clearance of most nanoparticles is through the liver. Thus, it is not surprising to see positive results using this model because both particles would eventually accumulate in the liver. However, it is hard to see how this nanoparticle system can work in any model other than hepatic models when they are administered systemically.
3. In the new supplementary Figure 22, it is obvious the different biodistributions of the two nanoparticles: for example, one accumulates in the heart and the other does not. This data further supports my previous point that the two nanoparticles would have different biodistributions due to their different sizes.

Point-by-point Response to Reviewers' Comments (NCOMMS-23-37148A)

Reviewer #1

Comments:

The authors have addressed all concerns in the revised manuscript and attempted to clarify the major concern regarding the “off-on” signals of ALCNs. However, there are still several misleading texts in the manuscript that should be clarified for the readers before publication.

Response: We really appreciate and agree with the reviewer's constructive comments.

1. The authors explained the “off-on” signals of ALCNs. For the “off” signal, both AIN and ASN are negatively charged at pH 7.4, which blocks singlet oxygen transfer and thereby prevents afterglow luminescence generation. For the “on” signal, the charge of ASN will be positive while AIN remains negative, resulting in an electrostatic attraction that allows singlet oxygen transfer for downstream oxidation of afterglow substrate to form a dioxetane intermediate and generate afterglow luminescence. To avoid the misunderstanding of the “off-on” signal, I highly recommend the authors clarify this in the manuscript. For instance, the authors could add: NIR unresponsive “off” and NIR responsive “on”, or $^1\text{O}_2$ inactivated “off” and $^1\text{O}_2$ activated “on”, or anything that could better clarify this point. This should be stated very clearly in the text so that the readers will not misunderstand that the off-on signals can be manually controlled.

Response: Thanks for the reviewer's constructive comments. We have clarified the $^1\text{O}_2$ inactivated afterglow “off” and $^1\text{O}_2$ activated afterglow “on” in the text and Figure caption as the suggested by reviewers.

Revised Figure 1. **b** Schematic illustration of ALCNs for pH-activatable upconversion afterglow luminescence. Under physiological conditions, ALCNs are in the $^1\text{O}_2$ inactivated afterglow "off" state due to the electrostatic repulsion-mediated the block of $^1\text{O}_2$ transfer from the donor AIN to acceptor ASN. In contrast, acidic pH induces the charge reversal of ASN and mediates the proximity to AIN via electrostatic interaction, which facilitates $^1\text{O}_2$ transfer from AIN to ASN, ultimately inducing $^1\text{O}_2$ activated afterglow "on".

Page 2, left panel, line 18: Under physiological conditions, ALCNs are in the $^1\text{O}_2$ inactivated afterglow "off" state.

Page 2, right panel, line 8: Thereby, the proximity between AIN and ASN, mediated by electrostatic interaction, enables efficient $^1\text{O}_2$ transfer from AIN to ASN, ultimately inducing $^1\text{O}_2$ activated afterglow "on".

2. As the authors stated, the pH-responsive mechanism of ALCNs is irreversible due to the hydrolyzed acid-labile amide bond of ASN that liberates the amino-termini group. The authors should discuss this fact in the discussion and include that the "off-on" signals of ALCNs are influenced by the tumor microenvironment.

Response: We appreciate the reviewer's comments. We have added the relevant discussion in the revised manuscript.

Page 10, right panel, line 9 from the bottom: The acid-labile amide bond of ASN within ALCNs is hydrolyzed to liberate the amino-termini group in an acidic environment, which is an irreversible process. This induces the "off-on" afterglow being easily perturbed by the tumor environment and thus is not applicable in a complex and dynamic biological

environment. Therefore, advances of acid-activatable afterglow luminescence from irreversibility to reversibility will be further explored in the future study.

3. For question 5, the authors have answered the question regarding the afterglow intensity of ALCNs under different conditions in Supplementary Fig. 9. However, the afterglow duration of ALCNs under different conditions is still missing.

Response: We appreciate the reviewer's comments. We have carried out the experiment and added the relevant information to Supplementary Fig. 10.

Page 5, left panel, line 10 from the bottom: The afterglow half-life changed conversely with the afterglow intensity of ALCNs at different laser irradiation time, power density and the nanoparticle concentrations, which was probably relevant to the duration time of unstable peroxides at different conditions (Supplementary Fig. 10).

Supplementary Fig. 10. The afterglow duration of ALCNs under different conditions. a Afterglow luminescence attenuation curve of ALCNs (20 μ g/mL ASN and 2 μ g/mL AIN) after illumination by 808 nm laser at different power densities within 20 min in 1 \times HEPES buffer (pH 5.5). b Afterglow luminescence attenuation curve of ALCNs after lamination by 808 nm (1 W/cm²) laser as a function of the concentration within 20 min in 1 \times HEPES buffer (pH 5.5). c Afterglow luminescence attenuation curve of ALCNs (20 μ g/mL ASN and 2 μ g/mL AIN) at different light irradiation time by 808 nm laser at power density of 1 W/cm² within 20 min in 1 \times HEPES buffer (pH 5.5). d-e Corresponding half-life changes as a function of different conditions. The error bars represented the standard deviation ($n = 3$).

Reviewer #2

Comments:

In the revised manuscript, the authors have well addressed the comments raised. The new data collected have well improved the revised manuscript. The paper is suggested for the accept for the publication at current version.

Response: We really appreciate and agree with the reviewer's constructive comments.

Reviewer #3

Comments:

In this revised manuscript, the authors added some more data in an attempt to address some of the limitations of their nanoparticle system. Though the added data provided some clarifications, it is still unconvincing that the biggest limitations have been addressed. The two-particle system dictates the largest obstacle for this system is to bring the two particles close enough in vivo for singlet oxygen transfer to happen. It can only happen using specific animal models as the authors showed in their liver tumor model. This nanoparticle system lacks general application unless these limitations can be addressed. Here are my responses to the authors' revision:

Response: We really appreciate the opinions of the reviewers for bringing us in-depth brainstorming. In term of the issues including the limitation of system of two nanoparticles and specified in vivo applications, we have provided a detailed response in the following.

1. I appreciate that the authors provided more references stating the diffusion distance of singlet oxygen is around 10-55 nm in living cells. This means the two nanoparticles will have to come into at least this distance for the singlet oxygen transfer to happen. When the nanoparticles are administered systemically, it is very difficult to ensure this would happen.

Response: We appreciate the reviewer's comments. We admit that this strategy is relatively low efficient, which induces the compromised imaging performance. But owing to the ultralow background of afterglow luminescence, only a small tiny of donor and acceptor nanoparticles with opposite charges in the acidic environment gather together, which can obtain considerable imaging contrast in diseased sites (as shown in Figure 4-6 in the revised manuscript). Similarly, click reactions (e.g. azide-alkyne and copper-free iEDDA click reactions) using two different components have been widely validated and explored for imaging or therapy (*Nat Commun.* 2022, 13, 4318; *Nature.* 2020, 579, 421; *Angew Chem Int Ed.* 2020, 59, 7864; *Nat. Chem. Biol.* 2017, 13, 415). Such strategy generally

utilizes the gathering of two different functional components (small molecules or nanoparticles) via effective click reactions in target of interest to realize the accumulation of imaging or therapeutic agents in diseased sites.

The availability and broad applicability of our strategy have been further validated in another two mice models including B16F10-xenograft subcutaneous tumor and pulmonary metastasis models through the systemic administration of nanoparticles (Fig. 6 and Supplementary Fig. S26). We have added the relevant results in the revised manuscript.

Page 9, left panel, line 7:

Afterglow imaging of tumors with Bio-ALCNs

To further enhance the tumor targeting ability of ALCNs for exploring the wide applicability, a tumor targeted group (i.e., biotin) was modified on the nanoparticles to obtain Bio-ALCNs (Fig. 6a). Compared with ALCNs, Bio-ALCNs possessed comparable size distribution and consistent afterglow responsivity towards acidic pH (Supplementary Fig. 25). Bio-ALCNs were firstly tested in biotin receptor-overexpressed B16F10 tumor-bearing xenograft mice^{62, 63}. After systemic administration of Bio-ALCNs, a gradually enhanced afterglow signal was observed, which reached the maximum at 12 h post-injection of nanoparticles (Supplementary Fig. 26). At this time point, the afterglow SNR of Bio-ALCNs-treated mice was 2.5- and 9.5-fold higher than that of ALCNs- and C-ALCNs-treated mice, showing the superior tumor targeting and responsivity of Bio-ALCNs (Supplementary Fig. 26). Fluorescence imaging also demonstrated the high tumor accumulation capacity of nanoparticles after biotin modification, but endowing obviously decreased SNR due to its high autofluorescence (Supplementary Fig. 26). Ex vivo imaging of resected tumor tissues, H&E analysis, and nanoparticles co-localization via fluorescence imaging of tumor sections further demonstrated the activated afterglow signal was actually originated from the activatable afterglow signal of Bio-ALCNs resided in the tumor tissues (Supplementary Fig. 26-28). The higher fluorescence signal in the liver relative to the other groups was probably derived from the metabolization of nanoparticles in the tumors via the liver tissue (Supplementary Fig. 28).

After demonstrating the imaging capability of subcutaneous xenograft tumor model via systemic administration, Bio-ALCNs were further investigated for imaging of pulmonary metastases in living mice (Fig. 6). The afterglow luminescence and fluorescence signals were continuously acquired from 0 to 48 h after tail vein injection of nanoparticles (Fig. 6a). As shown in Fig. 6b, an obvious afterglow signal in the lung site emerged at 8 h and reached the peak at 24 h post-injection of Bio-ALCNs. Owing to the enhanced targeting efficiency after biotin modification, the SNR of Bio-ALCNs was as high as 410.0, which was 3.7-fold higher than that of ALCNs (Fig. 6b, d). In comparison, an ignorable afterglow signal

was observed in the acidity-insensitive C-ALCNs-treated control group, confirming the pH-responsivity ability of Bio-ALCNs. Surprisingly, at all the observable time window, the suspected lesion in the lung site could not be clearly delineated in fluorescence imaging. The data confirmed that the ultrahigh sensitivity and the superior specificity of pH-activatable Bio-ALCNs for imaging of pulmonary metastases in afterglow imaging as compared with fluorescence imaging (Fig. 6d, e). Biodistribution study showed consistent results even at 48 h post-injection of nanoparticles (Fig. 6c, f, Supplementary Fig. 29). Moreover, the H&E assay and obvious co-localization of ASN and AIN through fluorescence imaging of the suspected lesion further supported the specific detection of pulmonary metastatic tumors by Bio-ALCNs (Fig 6g, Supplementary Fig. 30). Taken together, as a representative trial, this result proved the great targeting ability, sensitivity, and specificity of Bio-ALCNs for tumor imaging and also implied the potential broad utility of ALCNs for tumor detection with diverse design flexibility.

Fig. 6 | In vivo activatable afterglow imaging of pulmonary metastasis tumors. a Schematic illustration of the detailed procedures of afterglow imaging of 4T1 pulmonary

metastasis tumors. **b** Afterglow images under an 808 nm pre-irradiation (left panel), and fluorescence images (right panel) after intravenous injection of Bio-ALCNs, ALCNs or C-ALCNs (4 mg/kg Bio-ASN, ASN or C-ASN, and 0.4 mg/kg Bio-AIN, or AIN). The fluorescence images were acquired with an excitation wavelength at 480 nm and emission wavelength at 580 nm. **c** Ex vivo afterglow images of various tissues under an 808 nm laser pre-irradiation from 4T1 pulmonary metastasis-bearing mice at 48 h post-injection of Bio-ALCNs, ALCNs or C-ALCNs intravenously. **d, e** The quantified SNRs for afterglow luminescence imaging under 808 nm pre-irradiation (d), fluorescence imaging (e) of 4T1 pulmonary metastasis tumor-bearing mice as a function of time. Statistical analysis was performed with two-way ANOVA ($n = 3$, ns: not statistically significant, $***p < 0.001$). **f** Quantification of ex vivo afterglow intensities of various tissues in Fig. 6c. Statistical analysis was performed with one-way ANOVA ($n = 3$, ns: not statistically significant, $***p < 0.001$). **g** H&E staining images of tumor slices of mice after treatment with Bio-ALCNs, ALCNs or C-ALCNs (N: normal tissue, T: tumor tissue). The scale bar represents 100 μm . The error bars represent the standard deviation.

Supplementary Fig. 26. In vivo activatable afterglow imaging of B16F10-xenograft tumors. **a** Afterglow images under an 808 nm pre-irradiation (left panel), and fluorescence images (right panel) after intravenous injection of Bio-ALCNs, ALCNs or C-ALCNs (4 mg/kg Bio-ASN, ASN or C-ASN, and 0.4 mg/kg Bio-AIN, or AIN). The fluorescence images were

acquired with an excitation wavelength at 480 nm and emission wavelength at 580 nm. **b** Ex vivo afterglow images of various tissues under an 808 nm laser pre-irradiation from B16F10 tumor-bearing mice at 48 h post-injection of Bio-ALCNs, ALCNs or C-ALCNs intravenously. **c**, **d** The quantified SNRs for afterglow luminescence imaging under 808 nm pre-irradiation (c), or fluorescence imaging (d) of B16F10-tumor-bearing mice as a function of time. Statistical analysis was performed with two-way ANOVA ($n = 3$, ns: not statistically significant, $***p < 0.001$). **e** Quantification of ex vivo afterglow intensities of various tissues in b. Statistical analysis was performed with one-way ANOVA ($n = 3$, ns: not statistically significant, $***p < 0.001$). **f** H&E staining images of tumor slices of mice after injection with Bio-ALCNs, ALCNs or C-ALCNs (N: normal tissue, T: tumor tissue). The scale bar represents 100 μm . The error bars represent the standard deviation.

Supplementary Fig. 27. Study of biodistribution. a Ex vivo fluorescence images of various organs and tissues from B16F10 tumor-bearing mouse after intravenous injection of Bio-ALCNs, ALCNs, and C-ALCNs (4 mg/kg ASN or C-ASN, and 0.4 mg/kg AIN) at 48 h with an excitation at 480 nm and emission at 580 nm. b Quantification of fluorescence intensities of organs and tissues. The error bars represent the standard deviation ($n = 3$, ns: not statistically significant).

Supplementary Fig. 28. Fluorescence imaging of B16F10-xenograft tumor sections. Fluorescent images of B16F10 subcutaneous tumor sections after treatment with Bio-ALCNs, ALCNs or C-ALCNs scanning by Digital pathology scanner. ASN (green, 4 mg/kg, with an excitation at 490 nm and emission at 570 ± 10 nm), AIN (red, 0.4 mg/kg, with an excitation at 750 nm and emission at 780 ± 10 nm), nucleus (blue, 10 μM , with an excitation at 350 nm and emission at 420 ± 10 nm). The scale bar represents 10 μm .

Supplementary Fig. 29. Study of biodistribution. **a** Ex vivo fluorescence images of various organs and tissues from pulmonary metastasis tumor-bearing mouse after intravenous injection of Bio-ALCNs, ALCNs and C-ALCNs (4 mg/kg Bio-ASN, ASN, or C-ASN, and 0.4 mg/kg AIN) at 48 h with an excitation at 480 nm and emission at 580 nm. **b** Quantification of fluorescence intensities of organs and tissues. The error bars represent the standard deviation ($n = 3$, ns: not statistically significant).

Supplementary Fig. 30. Fluorescence imaging of 4T1 pulmonary metastasis tumor sections. Fluorescent images of 4T1 pulmonary metastasis tumor sections after treatment with Bio-ALCNs, ALCNs or C-ALCNs scanning by Digital pathology scanner. Bio-ASN, ASN (green, 4 mg/kg, with an excitation at 490 nm and emission at 570 ± 10 nm), Bio-AIN, AIN (red, 0.4 mg/kg, with an excitation at 750 nm and emission at 780 ± 10 nm), nucleus (blue, 10 μ M, with an excitation at 350 nm and emission at 420 ± 10 nm). The scale bar represents 10 μ m.

2. *In addition to the previously provided subcutaneous tumor model with intratumoral injection of the nanoparticles, the authors provided an orthotopic liver tumor model with systematic administration of nanoparticles. The orthotopic hepatic tumor model is not a very convincing model to use in this case since clearance of most nanoparticles is through the liver. Thus, it is not surprising to see positive results using this model because both particles would eventually accumulate in the liver. However, it is hard to see how this nanoparticle system can work in any model other than hepatic models when they are administered systemically.*

Response: We appreciate the reviewer's comments. To further explore the wide applicability of this strategy, Bio-ALCNs were prepared with higher tumor targeting ability by modifying the targeting group (i.e., biotin) on the nanoparticle surface of ALCNs. Attributing to the ultrahigh sensitivity of afterglow and high targeting efficiency of nanoparticles, Bio-ALCNs have been successfully used for afterglow imaging in another

two mice models including B16F10-xenograft subcutaneous and pulmonary metastasis tumor models through the systemic administration of nanoparticles (Fig. 6 and Supplementary Fig. S26). Therefore, as a representative trial, this result proved the great targeting ability, sensitivity, and specificity of Bio-ALCNs for tumor imaging and also implied the potential broad utility of ALCNs for tumor detection with diverse design flexibility.

3. In the new supplementary Figure 22, it is obvious the different biodistributions of the two nanoparticles: for example, one accumulates in the heart and the other does not. This data further supports my previous point that the two nanoparticles would have different biodistributions due to their different sizes.

Response: Thanks for the reviewer's comments. We really admit that the nanoparticles with different biodistribution behavior hampered the imaging performance. As we have stated in response to the comment 1, owing to the ultralow background of afterglow luminescence, only the gathering of a small tiny of nanoparticles could obtain considerable imaging contrast in diseased sites. The biorthogonal reaction always adopts the in vivo proximity to initiate the click reaction between two distinct molecules or nanoparticles with different biodistribution behavior to achieve imaging or therapy aims (*Nat Commun.* 2022, 13, 4318; *Nature.* 2020, 579, 421; *Angew Chem Int Ed.* 2020, 59, 7864; *Nat. Chem. Biol.* 2017, 13, 415). The data in Figure 4-6 also proved the feasibility of our strategy.

Reviewers' Comments:

Reviewer #1:

Remarks to the Author:

My concerns have now been adequately addressed. I'm happy to recommend its publication as is.

Reviewer #3:

Remarks to the Author:

In the revised manuscript, the authors improved the targeting ability of the two-particle system by introducing biotin as a targeting moiety to both particles and demonstrated the enhanced targeting ability in the melanoma xenograft model and pulmonary metastases model. This strategy could address most of my previous concerns. I recommend this manuscript to be published providing that the authors include detailed discussions of the potential limitations of this system in the discussion section.

Point-by-point Response to Reviewers (NCOMMS-23-37148B)

Reviewer #1

Comments:

My concerns have now been adequately addressed. I'm happy to recommend its publication as is.

Response: We really appreciate and agree with the reviewer's comments.

Reviewer #3

In the revised manuscript, the authors improved the targeting ability of the two-particle system by introducing biotin as a targeting moiety to both particles and demonstrated the enhanced targeting ability in the melanoma xenograft model and pulmonary metastases model. This strategy could address most of my previous concerns. I recommend this manuscript to be published providing that the authors include detailed discussions of the potential limitations of this system in the discussion section.

Response: Thanks for the reviewer's constructive comments. We have added the relevant discussion accordingly.

Page 11, line 2: Given that the donor and acceptor nanoparticles within ALCNs possess inconsistent physiological circulation and biodistribution, the gathering efficiency of nanoparticles within tumors is low, resulting in compromised imaging performance⁶⁷. Though Bio-ALCNs with enhanced tumor targeting ability present better imaging capacity relative to ALCNs, it can be only adapted to the tumors with overexpressed biotin receptor^{68,69}. Therefore, it is deserved to develop new strategies in terms of precise nanoengineering of nanoparticles to increase the tumor accumulation efficiency of the nanoparticles within diseased sites as well as broaden the applicability in different diseases^{70,71}.